# Developing a new cleavable crosslinker reagent for in-cell crosslinking
Fränze Müller [1,7], Bogdan R. Brutiu [2,7], Iakovos Saridakis[2], Thomas Leischner[2], Micha J. Birklbauer [3,4], Manuel Matzinger [1], Mathias Madalinski[1,5,6], Thomas Lendl[1], Saad Shaaban[2], Viktoria Dorfer [3] ✉, Nuno Maulide [2] ✉ & Karl Mechtler [1,5,6] ✉

Crosslinking mass spectrometry (XL-MS) is a powerful technology that recently emerged as an essential complementary tool for elucidating protein structures and mapping interactions within a protein network. Crosslinkers which are amenable to post-linking backbone cleavage simplify peptide identification, aid in 3D structure determination and enable system-wide studies of protein-protein interactions (PPIs) in cellular environments. However, state-of-the-art cleavable linkers are fraught with practical limitations, including extensive evaluation of fragmentation energies and fragmentation behavior of the crosslinker backbone. We herein introduce DiSPASO (bis(2,5-dioxopyrrolidin-1-yl) 3,3'-((5-ethynyl-1,3-phenylene)bis(methylenesulfinyl))dipropanoate) as a lysine-selective, MS-cleavable crosslinker with an alkyne handle for affinity enrichment. DiSPASO was designed and developed for efficient cell membrane permeability and crosslinking while securing low cellular perturbation. We tested DiSPASO employing three different copper-based enrichment strategies using model systems with increasing complexity (Cas9-Halo, purified ribosomes, live cells). Fluorescence microscopy in-cell crosslinking experiments revealed a rapid uptake of DiSPASO into HEK 293 cells within 5 minutes. While DiSPASO represents progress in cellular PPI analysis, its limitations and low crosslinking yield in cellular environments require careful optimisation of the crosslinker design, highlighting the complexity of developing effective XL-MS tools and the importance of continuous innovation in accurately mapping PPI networks within dynamic cellular environments.

Crosslinking mass spectrometry (XL-MS) has seen significant advancements over the past few decades, evolving into a powerful complementary technique for studying the spatial arrangement of proteins and their interactions within complexes[1–5]. XL-MS is particularly useful for studying protein structures and interactions in their native environments, including cells, organelles, and tissues[1,2,6] and has therefore significantly enhanced structural analysis by offering precise distance constraints, intrinsic to their chemical properties[7]. This capability is particularly valuable for resolving the three-dimensional architecture of proteins and protein complexes, especially when combined with complementary techniques such as HDX-MS or cryo-EM[3,7–11].

MS-cleavable crosslinkers, a class of crosslinkers amenable to backbone fragmentation upon MS acquisition, have gained attention in XL-MS by greatly simplifying the MS analysis and facilitating the identification of crosslinked peptides[12–17]. The data analysis process is supported by the generation of characteristic doublets of fragment ions during MS2 fragmentation, eventually enabling straightforward and accurate identification of crosslinks even in complex mixtures. Moreover, cleavable crosslinkers have enabled system-wide XL-MS studies, capturing protein-protein interactions (PPIs) in cellular environments, including weak or transient interactions[18–21]. This advancement allows for the exploration of the structural dynamics of proteins in their native state. Despite advancements in

[1]Institute of Molecular Pathology (IMP), Vienna BioCenter (VBC), Campus-Vienna-Biocenter 1, 1030 Vienna, Austria. [2]Institute of Organic Chemistry, University of Vienna, Währinger Straße 38, 1090 Vienna, Austria. [3]Bioinformatics Research Group, University of Applied Sciences Upper Austria, Softwarepark 11, 4232 Hagenberg, Upper Austria, Austria. [4]Institute for Symbolic Artificial Intelligence, Johannes Kepler University Linz, Altenberger Straße 69, 4040 Linz, Upper Austria, Austria. [5]Institute of Molecular Biotechnology (IMBA), Austrian Academy of Sciences, Vienna BioCenter (VBC), Dr. Bohr-Gasse 3, 1030 Vienna, Austria. [6]Gregor Mendel Institute (GMI), Austrian Academy of Sciences, Vienna BioCenter (VBC), Dr. Bohr-Gasse 3, 1030 Vienna, Austria. [7]These authors contributed equally: Fränze Müller, Bogdan R. Brutiu. ✉e-mail: viktoria.dorfer@fh-hagenberg.at; nuno.maulide@univie.ac.at; karl.mechtler@imp.ac.at

https://doi.org/10.1038/s42004-025-01568-1                                                                **Article**

data analysis, there remain challenges in accurately determining false discovery rates (FDR) for identified PPIs. Furthermore, the complexity of the data requires careful handling of error estimation to ensure the reliability of reported PPIs[22,23]. Additionally, there are practical limitations such as the requirement for specific conditions for cleavage, the potential for incomplete cleavage, and the need for specialized mass spectrometry setups to detect and analyze cleavage products effectively[24]. Hence, crosslinkers are often combined with enrichable groups, such as biotin[25–29], known for its high affinity towards streptavidin or avidin, and sometimes with MS-cleavable groups, for proteome-wide analyses[30–32]. However, the strong streptavidin-biotin interaction can raise difficulties upon release from the solid support, prompting alternative strategies like solid-supported monomeric avidin or incorporating release groups in the crosslinker backbone[28,33,34]. Various release groups, including PEG[25], pinacol esters[25,30], azobenzenes[27], disulfides[27,35], and photocleavable groups[27], have been explored. Click chemistry groups offer versatility, serving as capture or enrichment groups, with reversible options like acid-cleavable acetal or disulfide bonds[35–41]. While this innovation marks significant progress, it also presents opportunities for further refinement, as demonstrated by the development of a negatively charged phosphonic acid as a crosslinker reagent[42]. While this offers simplicity, its negative charge inhibits cell membrane crossing, hindering in-cell crosslinking. This hurdle could be overcome by protecting the negative charge with a chemical protection group[43]. Chemical modifications in crosslinking reagents offer promising functionalities but also pose challenges like increased hydrophobicity. Cleavable crosslinkers have advanced XL-MS, enhancing protein structure analysis, though further improvements are needed to overcome current limitations.

To overcome the aforementioned limitations, we hereby introduce DiSPASO (**1**, full name: bis(2,5-dioxopyrrolidin-1-yl) 3,3'-((5-ethynyl-1,3-phenylene)bis(methylenesulfinyl))dipropanoate) as a novel lysine reactive, MS-cleavable and membrane-permeable crosslinker featuring an alkyne-based click chemistry handle for affinity enrichment. This study provides a detailed exploration of the design, synthesis, and careful characterization of DiSPASO, providing a comprehensive evaluation of its chemical properties, effectiveness in mapping protein-protein interactions, and performance relative to existing crosslinkers. This analysis will also explore DiSPASO's fragmentation behavior in mass spectrometry, highlighting both its advantages and the challenges encountered, thereby offering insights into its potential and areas for further optimization in crosslinking mass spectrometry applications.

## Results
### Synthesis of a novel cleavable and enrichable crosslinker reagent DiSPASO and its modular perspective
We designed the crosslinker DiSPASO (**1**) bearing four key elements: a stable underexplored core decorated with an enrichment site (terminal alkyne), two side-chains containing the cleavable sulfoxide moieties and terminal NHS-esters for targeting lysins. A retrosynthetic analysis of compound (**1**) suggested a common intermediate **S5** in the synthesis pathway. **S5** was efficiently synthesized from commercially available diester amine **S1** through a series of reactions: a Sandmeyer reaction to introduce the aryl bromide, reduction of the ester groups, an Appel-type bromination and finally substitution with the corresponding methyl ester-bearing thiol. **S5** was obtained on a gram scale. The aryl bromide moiety presents an ideal candidate for cross-coupling with various enrichment sites. In this study, we chose a Sonogashira coupling. A global deprotection to the diacid, followed up by NHS coupling formed precursor **S8a**. Optimized oxidation delivered the novel enrichable crosslinker DiSPASO (**1**) in over 150 mg (Fig. 1A). With a streamlined synthesis in hand, the NHP derivative (**2**) was also obtained after two simple steps (Fig. 1B). The high-yielding steps can easily be scaled up.

### DiSPASO in context to other in-cell and lysate crosslinkers
DiSPASO was engineered as an in-cell crosslinker reagent with good cell membrane permeability properties while maintaining the solubility of the reagent during the in-cell crosslinking procedure. For a direct comparison of

DiSPASO in the context of other state-of-the-art crosslinkers used in in-cell or cell lysate crosslinking, we used a topological polar surface area (tPSA) against partition coefficient (cLogP) plot (Fig. 2). This visualization offers critical insights into the chemical properties of compounds, essential for design and optimization purposes. It effectively showcases the delicate balance between hydrophilicity and lipophilicity, crucial for understanding a compound's permeability across biological membranes. Compounds exhibiting lower tPSA values and higher cLogP values tend to display enhanced membrane permeability, owing to their greater affinity for lipid bilayers. DiSPASO shows medium membrane permeability and hydrophobicity properties compared to other crosslinker reagents and is in line with the BSP crosslinker published in Gao et al.[36,37].

Trifunctional crosslinkers, such as DiSPASO, enhance crosslinking mass spectrometry investigations by boosting sensitivity and specificity, primarily through an affinity tag that aids in enriching low-abundant crosslinked peptides that are typically masked by abundant linear peptides. This advancement is crucial for uncovering elusive protein interactions and enriching XL-MS data depth. DiSPASO's effectiveness in cellular entry and membrane permeation was evaluated using three different enrichment strategies in increasing sample complexities starting from single peptide crosslinking towards live HEK cell crosslinking.

### Comparison of DSBSO and DiSPASO crosslinking using Cas9-Halo
We optimized the affinity enrichment by assessing the following strategies (Fig. 3). The first enrichment employing picolyl azide (Fig. 3A, (**3**)) as a click handle and immobilized metal affinity chromatography (IMAC) for crosslinked peptide enrichment was tested using a single synthetic peptide (Ac-WGGGGRKSSAAR-COOH) with a defined crosslink-site (Supplementary Fig. S1A) and additionally using Cas9-Halo as the model protein (Supplementary Fig. S2A) to ensure a fully controlled environment. Both experiments demonstrated maximal crosslinked peptides/intensity when using 5 mM picolyl azide, with a click reaction efficiency of 99.3%. Click reaction efficiency was monitored by converting crosslinked products from DiSPASO crosslinking (Supplementary Fig. S2A, blue bars) to products formed after click reaction with picolyl azide (Supplementary Fig. S2A, orange bars), completing the click reaction when all DiSPASO links converted to click product BPNP (**4**) in Fig. 3A.

This principle was applied to all three enrichment strategies, tracking efficiency by monitoring mass shifts after each reaction's conversion. Following click reaction optimization, the optimal higher-energy collision dissociation (HCD) was determined via single peptide crosslinking. The best-combined score for a crosslinked peptide was achieved with an HCD of 34 (Supplementary Fig. S1B), despite the main doublet intensity decreasing with higher HCDs (Supplementary Fig. S1C). A balance between peptide backbone fragmentation and doublet intensity required for crosslink identification was essential. Consequently, optimal fragmentation energies of 25, 27, and 32 were selected for further experiments.

The fragmentation pattern of DiSPASO was evaluated on Cas9 crosslinking data (Fig. 4). Six potential fragments for the DiSPASO crosslinker were calculated based on the substitutions of the structures shown in Fig. 4A. The full crosslinker mass was also considered as fragment 7 to account for the theoretical cleavage of NH-crosslinker/peptide bounds. Fragment ETFP (**9**), ETHMP (**10**) and alkene (**11**) represent the common and expected fragmentation pattern that is known from several fragmentation studies regarding DSBSO[30,44] and DSSO[20,45,46]. Additionally, a second theoretical fragmentation pattern was tested due to low identification rates using DiSPASO for Cas9 crosslinking. The second fragmentation pattern includes a sulfenic acid fragment (**13**) or a thiol fragment (**14**) on the short side and an unexpected fragment EMP (**12**) on the long side of the crosslinker. Each fragment was defined separately according to the monoisotopic masses in Fig. 4D, E. Possible doublet distances were calculated according to Eq. (1) in the "Methods" section "Software adjustments". This resulted in theoretical 15 doublet distances that can occur during fragmentation of DiSPASO. The same

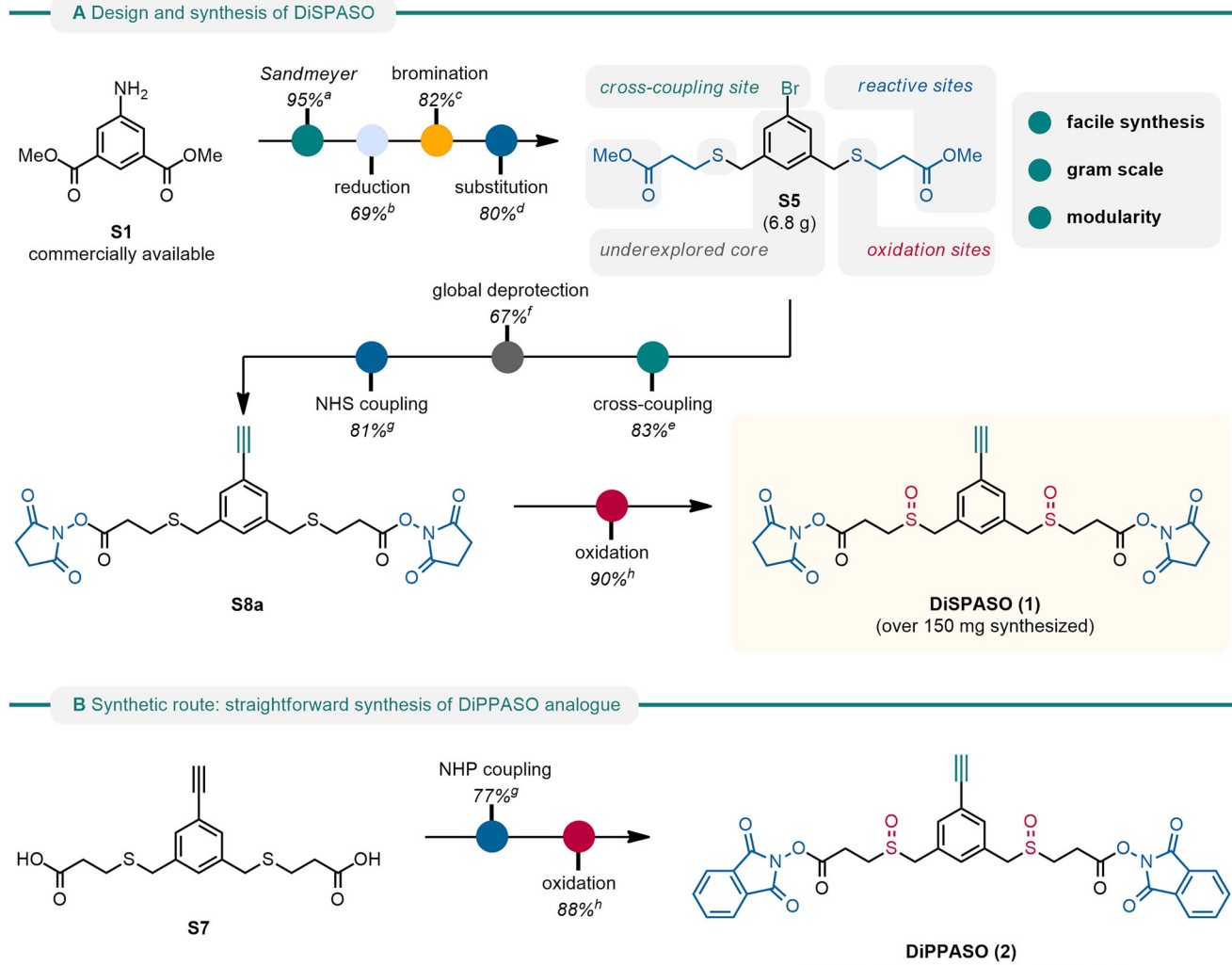

**Fig. 1 | Design and synthetic route of the DiSPASO and DiPPASO crosslinkers.**
**A** Design and synthesis of DiSPASO. **B** Synthetic route: straightforward synthesis of DiPPASO. [a]Sandmeyer Reaction: 1.2 eq. NaNO$_2$/HBr, 0 °C, 1 h, 1.4 eq. CuBr, 0 °C, 18 h. [b]Reduction: 2.0 eq. LiAlH$_4$, THF, 0 °C, 2 h. [c]Bromination: 8.5 eq. PBr$_3$, DCM, 40 °C, 15 h, then H$_2$O, 23 °C, 20 h. [d]Substitution: 2.05 eq. methyl 3- mercaptopropanoate, 2.05 eq. K$_2$CO$_3$, DMF, 40 °C, 30 h. [e]Cross-coupling: 1.0 eq. TMS-acetylene, 0.05 eq. Pd(dppf)Cl$_2$-CH2Cl2, 0.05 eq. CuI, THF, 60 °C, 20 h. [f]Global deprotection: 6.0 eq. LiOH, THF/H$_2$O (3:1), 23 °C, 20 h. [g]NHS/NHP coupling: 2.2 eq. NHS or NHP, 2.2 eq. EDCI-HCl, 0.2 eq. NEt$_3$, DMF, 23 °C, 24 h. [h]Oxidation: 2.0 eq. mCPBA, DCM, 0 °C, 2 h.

procedure was applied to DSBSO to compare the fragmentation behavior of both crosslinkers.

Azide-DSBSO shows similar chemistry on the crosslinker backbone since it has the same length and cleavage site as DiSPASO. The center of Azide-DSBSO also comprises a ring system but in contrast to DiSPASO it is not a conjugated pi-ring system and instead of an alkyne functionality DSBSO employs an azide group for the enrichment and click chemistry. Hence, DSBSO was considered an ideal positive control to compare to the fragmentation pattern that was already published before[30,45].

Surprisingly, DiSPASO showed various fragmentation sites with doublet D11 and D12 as the main occurring doublet distances. D11 represents the doublet distance between ETFP (9) and alkene (11) with a distance of 208 Da. The second main doublet is D12, ETHMP and alkene with a distance of 226.01 Da. Both doublets belong to the known fragmentation pattern published for DSSO before[34]. Additionally, DiSPASO shows doublets for the less likely fragmentation pattern between fragment EMP (12) and sulfenic acid (13) or thiol (14), D13 and D14 respectively, and alkene and thiol or sulfenic acid alone, D1 and D2 respectively. Whereas D13 and D14 represent 25% and 17% of the identified links, respectively, D1 and D2 account for 30% and 34%, highlighting the increased backbone fragmentation of the DiSPASO crosslinker. In contrast, DSBSO shows only

one main doublet distance D11, which is defined by a distance of 182 Da between the alkene fragment and the long thiol fragment of DSBSO. D11 resulted in 285 identifiable links on average. D1 and D5 represent the second most abundant doublet distances with 16% and 15%, respectively. Although DSBSO also shows additional doublet distances, the main doublet distance stands out compared to all others. In conclusion, DiSPASO shows a different fragmentation pattern, that yields an almost complete scattering of the crosslinker backbone, compared to DSBSO, although the main cleavage sites are the same in perspective of the backbone. Hence, for further DiSPASO analysis the main doublet distances D11 and D12 were employed for crosslink identifications in MS Annika.

With all optimized parameters in place, the picolyl azide enrichment yielded 312 unique residue pairs compared to 259 identified residue pairs in the non-enriched control sample (Fig. 5A). The picolyl enrichment increased the number of links by 17%, but the overall performance compared to DSBSO is lowered by 33% for 500 ng (695 links DSBSO, 465 links DiSPASO) and 29% for 200ng injection amount (363 links DSBSO, 259 links DiSPASO). Nevertheless, the technical reproducibility of DiSPASO crosslinks before and after enrichment is high with 37% and 47% overlap between triplicates, respectively (Fig. 5B). The recovery rate of crosslink before and after enrichment is 49% with 21% uniquely identified in the enriched sample.

**Fig. 2 | Topological polar surface area plot of commonly used crosslinker reagents.** DiSPASO is shown in the center of the plot (green dot) with moderate membrane permeability and hydrophobicity compared to other common crosslinker regents. DiPPASO (not presented in this study) shows similar properties to t-BuPhoX (TBDSPP) with a high hydrophobicity compared to other crosslinkers. The clouds of enrichable and permeable crosslinkers are clustered in two distinct parts of the plot, with DiSPASO at the interface between both groups. Classification of the crosslinkers was extracted from the Thermo Scientific crosslinker selection tool. If a crosslinker shows two properties of the classifications the more dominant property was selected for plotting.

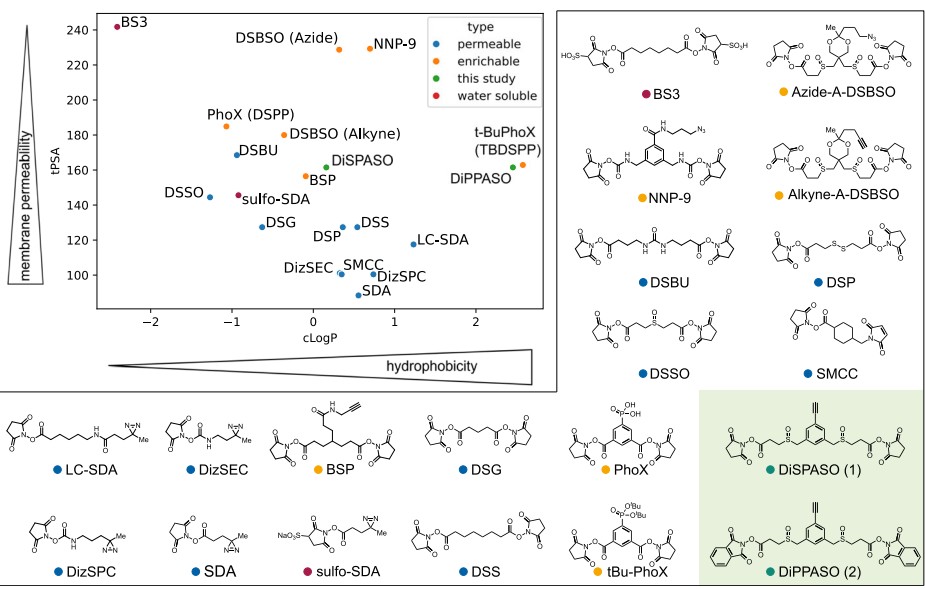

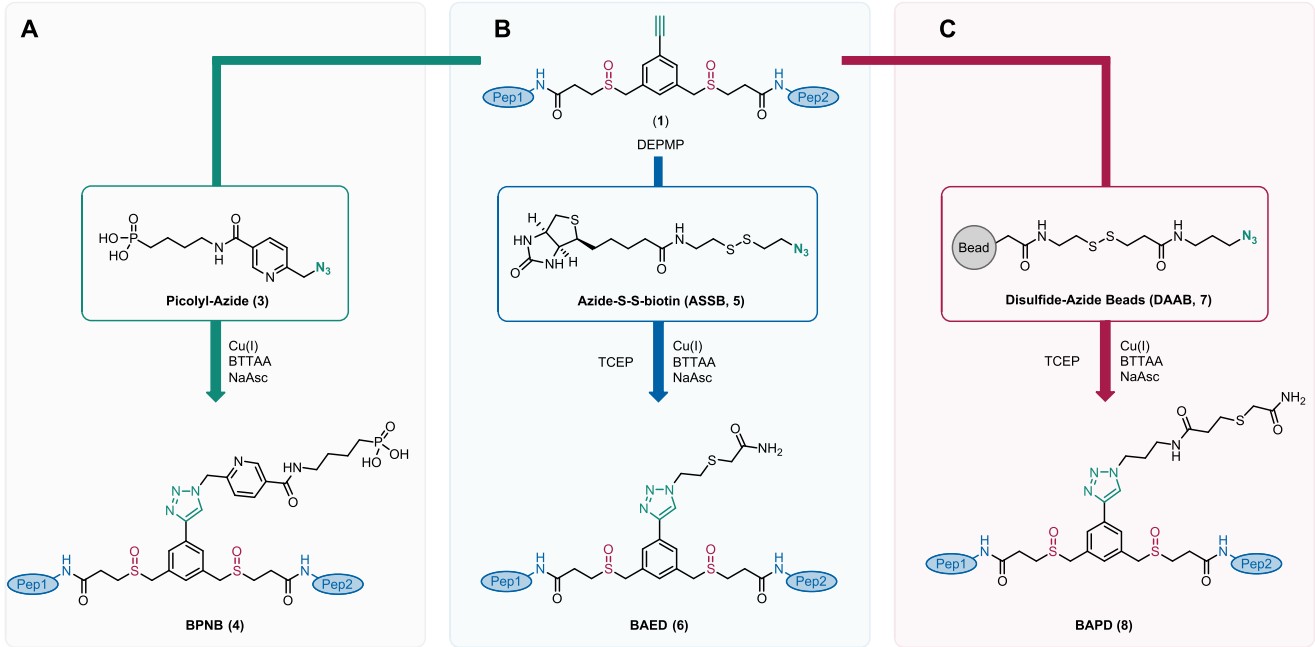

**Fig. 3 | Overview of DiSPASO enrichment strategies. A** DiSPASO enrichment using a picolyl azide as a click chemistry reagent and an IMAC-based enrichment strategy to retrieve crosslinked peptides from a complex mixture. **B** Azide-S-S-biotin (ASSB) based enrichment using ASSB as a click chemistry reagent and biotin-streptavidin bead strategy to enrich for crosslinked peptides. The elution of bead-bound crosslinked peptides is performed using the reduction of the disulfide bond of the ASSB compound. **C** Simplified version of (**B**). The click reagent Disulfide Azide is already coupled to the beads and the click reaction is taking place directly on the beads. Elution of crosslinked peptides is performed after the click reaction and washing procedure of the beads using a reducing reagent. IUPAC names of all compounds used in this manuscript are described in Supplementary Table S4.

The picolyl azide enrichment strategy was tested in a more complex system with crosslinked Cas9 as spike-in and HeLa cell lysate as background to increase the overall complexity of the sample (Fig. 6A and Supplementary Fig. S2B). The spike-in amount ranged from 0.5 ug to 10 ug crosslinked Cas9 in 100 ug HeLa lysate. The lower the spike-in amount the fewer crosslinks could be identified after enrichment. For 0.5 ug only 14 crosslinks and in the 1:100 ratio (near in-cell condition) 20 crosslinks of a triplicate injection could be identified. In-cell crosslinking experiments in HeLa cells were performed to prove the hypothesis that the picolyl azide enrichment strategy is underperforming in real-case scenarios. Indeed, for HeLa in-cell crosslinking only 1 crosslink could be identified (Fig. 6A, gray background).

To improve the enrichment of crosslinked peptides in in-cell crosslinking experiments we moved to a different enrichment strategy using an Azide-S-S-biotin click chemistry handle.

## Azide-S-S-biotin and Disulfide Azide Agarose bead enrichment

Azide-S-S-biotin (ASSB), known for its effectiveness in protein and peptide labeling and enrichment, was utilized to enhance crosslinked peptide enrichment in in-cell crosslinking, leveraging its cleavable disulfide bond for peptide release post-enrichment (Fig. 3B). Strong binding (free binding energy of −41.17 kcal/mol[47], $K_d$ of ~$10^{-30}$ M) of the biotin group to streptavidin beads allows efficient capture of labeled peptides, while elution efficiency can be

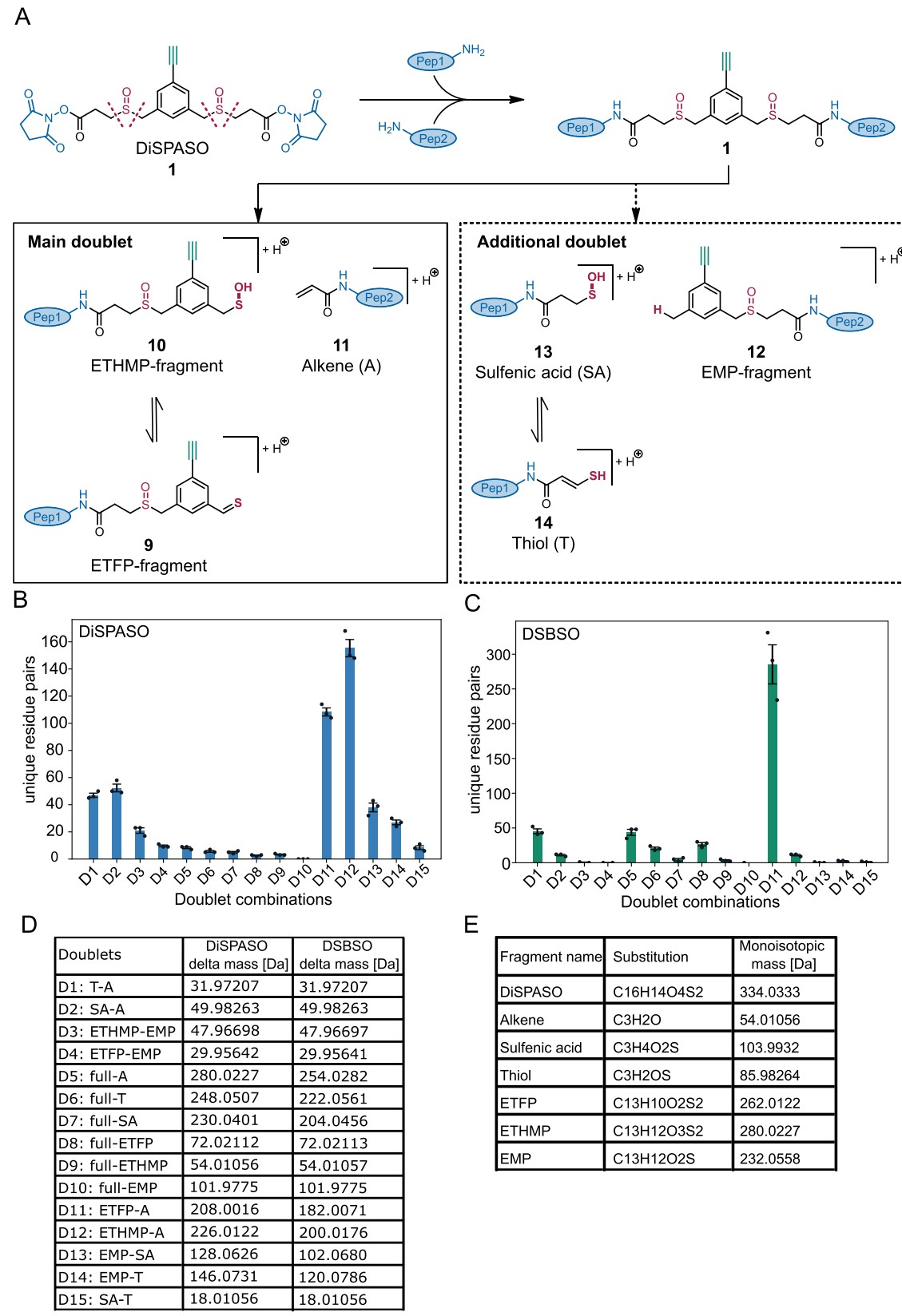

| Doublets | DiSPASO delta mass [Da] | DSBSO delta mass [Da] |
|---|---|---|
| D1: T-A | 31.97207 | 31.97207 |
| D2: SA-A | 49.98263 | 49.98263 |
| D3: ETHMP-EMP | 47.96698 | 47.96697 |
| D4: ETFP-EMP | 29.95642 | 29.95641 |
| D5: full-A | 280.0227 | 254.0282 |
| D6: full-T | 248.0507 | 222.0561 |
| D7: full-SA | 230.0401 | 204.0456 |
| D8: full-ETFP | 72.02112 | 72.02113 |
| D9: full-ETHMP | 54.01056 | 54.01057 |
| D10: full-EMP | 101.9775 | 101.9775 |
| D11: ETFP-A | 208.0016 | 182.0071 |
| D12: ETHMP-A | 226.0122 | 200.0176 |
| D13: EMP-SA | 128.0626 | 102.0680 |
| D14: EMP-T | 146.0731 | 120.0786 |
| D15: SA-T | 18.01056 | 18.01056 |

| Fragment name | Substitution | Monoisotopic mass [Da] |
|---|---|---|
| DiSPASO | C16H14O4S2 | 334.0333 |
| Alkene | C3H2O | 54.01056 |
| Sulfenic acid | C3H4O2S | 103.9932 |
| Thiol | C3H2OS | 85.98264 |
| ETFP | C13H10O2S2 | 262.0122 |
| ETHMP | C13H12O3S2 | 280.0227 |
| EMP | C13H12O2S | 232.0558 |

achieved due to reduction of the disulfide bond. The resulting enriched and cleaved crosslinked peptide is small in size, which ensures minimal impact on ionization efficiency during the electrospray ionization (ESI) process.

ASSB's optimization was conducted using the Cas9-Halo model, crosslinked with DiSPASO. Concentration titrations for ASSB were set from 1 mM to 20 mM, finding an optimal performance plateau at 10 mM with 175 unique residue pairs identified (Supplementary Fig. S3A). Despite varying ASSB concentrations, a consistent loss of crosslinked peptides was observed in the enrichment flowthrough, with a maximum loss of 333 links (Supplementary Fig. S3B). The concentration of sodium ascorbate (NaAsc),

**Fig. 4 | Possible fragments of DiSPASO during MS2 fragmentation and evaluation of MS Annika search setting regarding doublet distances.**
**A** Fragmentation products of DiSPASO after high-energy collision dissociation (HCD) showing structures of DSSO-like cleavage products (alkene 11, ETFP 9 and ETHMP-fragments 10, main doublet) and possible additional doublets of the long side of the crosslinker (EMP 12 with SA 13 or T 14 as stubs, dotted gray box).
**B** Numbers of residue pairs identified from Cas9 crosslinked with DiSPASO while searching with each doublet distance separately. D12 (ETHMP-alkene doublet) shows the maximum number of identified links with D11 (ETFP-alkene) as the second and D1, D2 (alkene-thiol, sulfenic acid-alkene) as the third abundant

doublets. **C** Numbers of residue pairs identified from Cas9 crosslinked with DSBSO while searching with each doublet distance separately. D11 (ETFP-alkene doublet) shows the maximum number of identified links. The number of technical replicates is indicated as separate black dots on top of the bar (*n* = 3). The standard deviation was estimated as average distance from each data point to the sample mean. **D** Table of doublet definitions and delta masses of all possible doublets. **E** Table of substitutions and monoisotopic masses of all fragments including DiSPASO as full construct bound to peptides. IUPAC names of all compounds used in this manuscript are described in Supplementary Table S4.

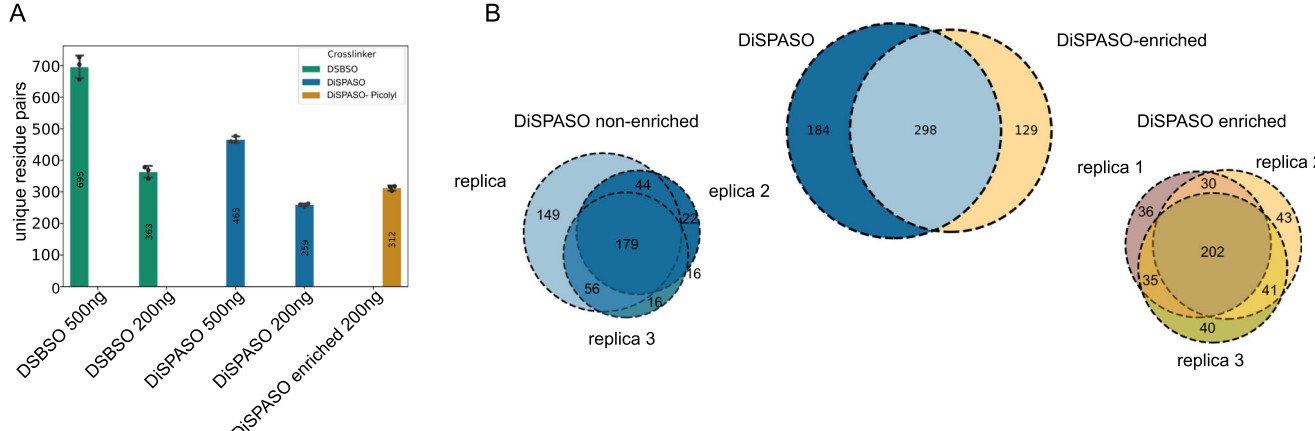

**Fig. 5 | Comparison of DiSPASO and DSBSO using Cas9 as a model protein, as well as enrichment performance of DiSPASO using picolyl azide as enrichment tag. A** Number of identified unique crosslinks (unique residue pairs) in a triplicate experiment of Cas9 crosslinked with either DiSPASO or DSBSO in different injection amounts (500 and 200 ng). Blue indicates non-enriched crosslinked peptides, yellow enriched crosslinked peptides from Cas9 DiSPASO experiments. The green color represents Cas9 crosslinking results using DSBSO. The mean value of residue pairs is plotted in the middle of the bar. The number of technical

replicates is indicated as separate black dots on top of the bar (*n* = 3). The standard deviation was estimated as average distance from each data point to the sample mean. **B** Overlap of identified crosslinks after click reaction and enrichment using picolyl azide as click chemistry reagent. After click reaction and enrichment using picolyl azide the numbers of crosslinks show an overlap of 49% with 21% unique to the enriched crosslinked sample. Left bottom of **B**: Overlap of technical replicates of non-enriched Cas9 links. Right bottom of **B**: Overlap of technical replicates of enriched Cas9 links.

crucial for efficient click reactions without compromising the disulfide bond in ASSB, was also optimized, reaching a peak efficiency at 20 mM NaAsc with 242 unique links (Supplementary Fig. S3C), but with a significant sample loss of 39% after enrichment. Adjusting NaAsc to 30 mM aimed to balance click reaction efficiency with minimal side reactions. Bead volume optimization for enrichment ranges from 10 μL to 100 μL of MBS bead slurry, with the highest number of detected links (217) at 70 μL (Supplementary Fig. S3E), though not entirely preventing sample loss. The sample loss could be reduced by 56% (158 links 20 mM ASSB vs. 95 links 70 uL bead slurry) but not avoided, even after using 100 uL of bead slurry (Supplementary Fig. S3F).

Various bead types were tested (Supplementary Fig. S4A, B) without markedly reducing sample loss, leading to the adoption of a third strategy using disulfide azide agarose beads (Fig. 3C), streamlining the workflow by clicking crosslinked peptides directly to the beads. No additional purification steps are needed to avoid unspecific binding by free ASSB molecules and therefore blocking binding sites on the beads. All approaches, despite the methodical optimization and strategic shifts, underscored the inherent challenges in achieving efficient and loss-minimized enrichment of crosslinked peptides.

### In-cell crosslinking using DiSPASO
The picolyl azide enrichment strategy was challenged by crosslinking live HeLa cells directly in a 6-well plate using the DiSPASO crosslinker. Unfortunately, this experiment resulted in only 1 crosslink although several attempts were made to push this workflow towards success (Fig. 6A, gray background).

Therefore, the ASSB enrichment workflow was applied to isolated commercial *E. coli* ribosomes in a complex HEK 293 cell background to

proceed with the crosslink enrichment in live cells (Supplementary Fig. S7A, B). The ASSB enrichment strategy resulted in 14 ribosomal crosslinks that could be enriched from a 1:100 mixture (1 ug crosslinked ribosome spike-in in 100 ug HEK 293 cell lysate). This result was not efficient enough for in-cell crosslinking experiments and hence, we tried to improve the enrichment procedure by employing Disulfide azide agarose beads to directly click the crosslinked peptides to the enrichment handle and the beads in one step. We here titrated the bead amount using crosslinked HEK 293 cells. The cells were crosslinked directly in a 10 cm dish using 5 mM DiSPASO resulting in approximately 9e6 crosslinked cells. The sample was split to each condition equally with a bead amount of 5–120 uL tested (Fig. 6C). Seventy-one unique crosslinks could be detected from the 120 uL bead sample, the maximum of crosslinks in this experiment. The linear peptide background could not be reduced across the conditions, possibly due to the unspecific binding of peptides to the beads which is common in bead-based enrichment strategies. The problem of high linear peptide background stayed constant across all tested enrichment strategies and experiments, even with extensive washing procedures, which pinpoints a systematic problem for this kind of enrichment workflow (Fig. 6B, D).

Despite the promising membrane permeability and solubility properties of DiSPASO, mass spectrometry experiments yielded low identification rates in in-cell crosslinking studies. To investigate this discrepancy, we proceeded with confocal microscopy to directly assess the crosslinker's performance in live cells on a visual basis.

### Membrane permeability assessment for DiSPASO in HEK cells
DiSPASO's performance for in-cell crosslinking was further evaluated through confocal microscopy, showing rapid uptake by HEK 293 cells, visible in cell nuclei within minutes (Fig. 7A and Supplementary Fig. S5).

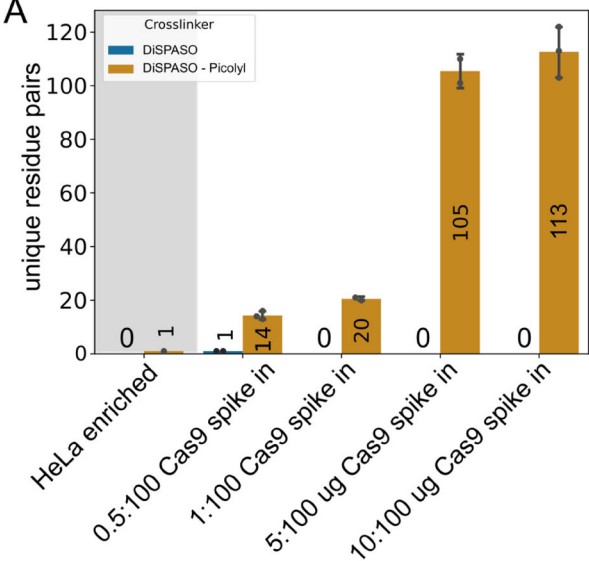

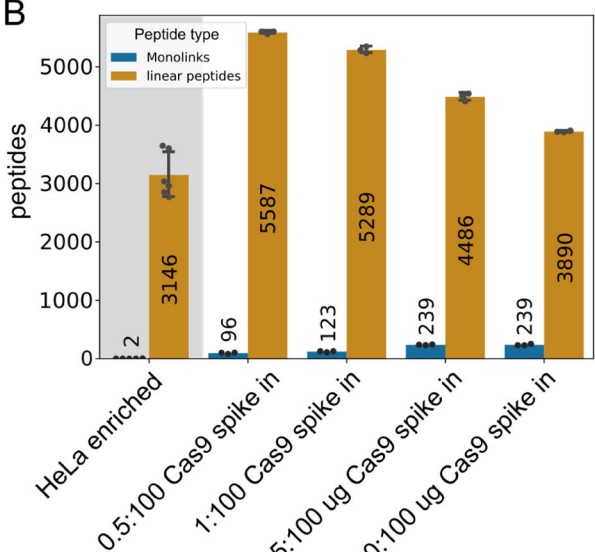

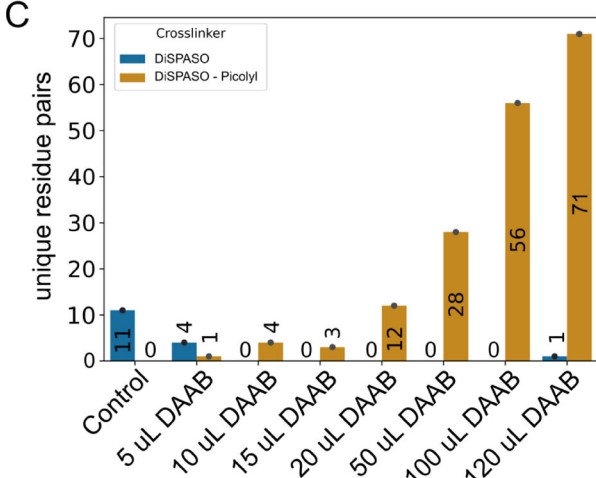

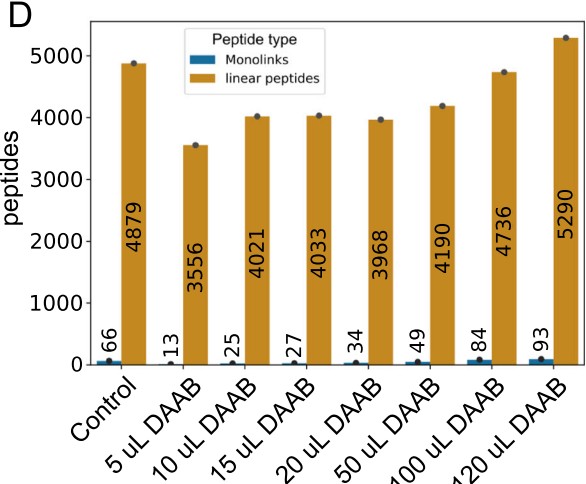

**Fig. 6 | Application of DiSPASO enrichment strategies in increasing sample complexity. A** Spike-in experiment of crosslinked Cas9 in HeLa background. The Cas9 spike-in amount increased from 0.5 ug to 10 ug in a constant background of 100 ug HeLa. A HeLa in-cell crosslinking sample is used as a "control" sample. The amount of enriched Cas9 links increases with the amount of Cas9 spike-in. **B** Analysis of Monolinks and linear peptides of the Cas9 spike-in experiment. The Monolinks and linear peptides of the "control" sample show in general fewer peptides due to the different experimental setup of in-cell crosslinking experiments in comparison to a spike-in experiment. The HeLa background peptides of the Cas9 spike-in decrease with increasing enrichable Cas9 crosslinks but cannot be depleted

completely. The number of technical replicates is indicated as separate black dots on top of the bar. The standard deviation was estimated as average distance from each data point to the sample mean. **C** HEK 293 in-cell crosslinking experiment using DiSPASO for crosslinking and Disulfide Azide Agarose beads (DAAB) for click chemistry-based enrichment of crosslinks. The bead amount is increased from 5 uL beads slurry to 120 uL, the control sample is crosslinked with DiSPASO without enrichment. With an increasing number of beads, the number of identifiable crosslinks after enrichment increases. **D** Analysis of Monolinks and linear peptides of the in-cell crosslinking experiment. Linear peptides could not be depleted after crosslink enrichment.

The HEK 293 cells, treated with 5 mM DiSPASO over 30 min, revealed a strong increase in signal within the first 5 min, reaching a plateau thereafter (Fig. 8A).

This indicates not only DiSPASO's adeptness at penetrating cellular membranes but also its quick reactivity once inside the cell. Such attributes are critical for effective in-cell crosslinking, enabling the capture of a broad spectrum of protein-protein interactions in their native cellular environment. The microscopy images serve as a direct visual affirmation of DiSPASO's capabilities, highlighting the controversy between the visual and the mass spectrometry results. To also evaluate the crosslinker performance compared to other commonly used in-cell crosslinking chemistry, we employed Azide-DSBSO[30,40] as a benchmark reactant. For a fair comparison, HEK 293 cells were treated with 5 mM DSBSO in the same manner and parallel to the DiSPASO treatment. The difference in the click handle resulted in the use of

two separate fluorescence dyes for each crosslinker. To label DiSPASO, the green dye AlexaFluor 488 and for DSBSO the red dye AlexaFluor 555 was used to visualize the performance of in-cell crosslinking (Fig. 7B). In contrast to DiSPASO, which enters the cell very quickly, DSBSO needs longer incubation times and reaches its maximum performance after 30 min (Fig. 8C). The extended reaction time may be attributed to the presence of the charged azide functionality incorporated into the crosslinker that decreases the membrane permeability of the crosslinker as shown in Fig. 2.

Our microscopy data, which validate the predicted chemical properties of DiSPASO, illustrate the crucial balance between hydrophilicity and lipophilicity that governs a compound's permeability across biological membranes. While these visual confirmations align with theoretical expectations, they also highlight the challenges encountered during sample preparation and mass spectrometry analysis, revealing the complexity of

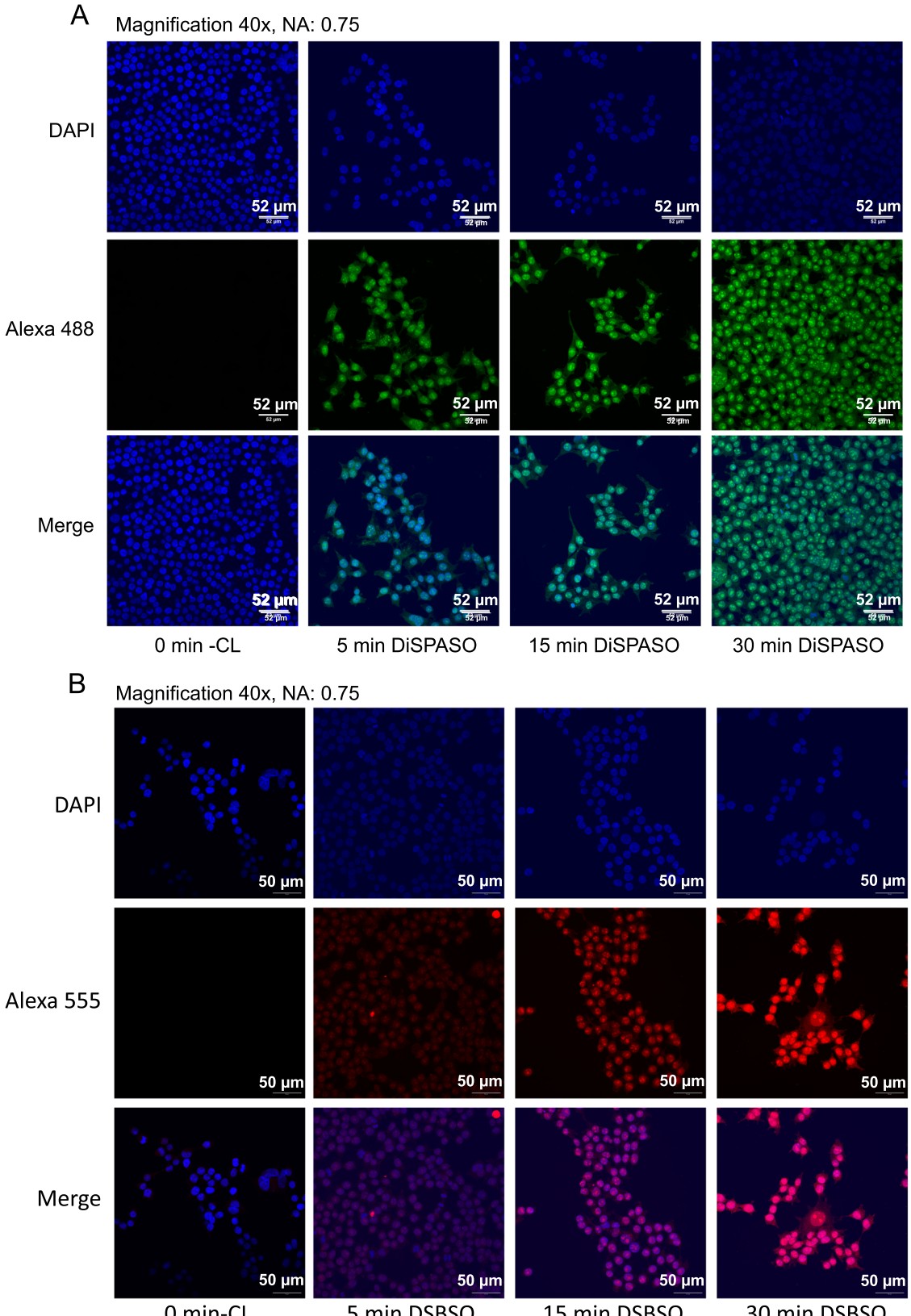

A  Magnification 40x, NA: 0.75

B  Magnification 40x, NA: 0.75

translating theoretical advancements into practical applications. We hypothesize that the ability of the crosslinker to diffuse throughout the cell and crosslink proteins in all cellular compartments may depend on the hydrolysis of one reactive NHS-ester group, which increases its solubility.

This could explain the observed discrepancy between the fluorescence signal, where the remaining NHS-ester reacts with proteins, and the low overall crosslinking yield in the mass spectrometry data, as indicated by the emergence of monolinks.

**Fig. 7 | Comparison of DiSPASO and DSBSO uptake during in-cell crosslinking experiments in HEK 293 cells. A** Confocal microscopy images of DiSPASO during in-cell crosslinking experiments with a crosslink duration of 0 min (Control sample without crosslinking, first left panel), 5 min (left second panel), 15 min (third left panel), 30 min (last panel). The nuclei fluorescence signal of DAPI is shown in the upper panel in blue, fluorescence of crosslinked proteins after click reaction to Alexa 488 (green) in the middle panel and a merge of both channels on the bottom. The images were taken on an Olympus Spinning Disk Confocal microscope (2-024) using a magnification of 40 and a numerical aperture of 0.75. **B** In comparison the

DSBSO in-cell experiments were performed in the same way except for the fluorophore. DSBSO has an azide as click reaction handle and therefore Alexa 555 (red) was used to visualize crosslinked proteins. The crosslink duration was set to 0 min (Control sample without crosslinking, first left panel), 5 min (left second panel), 15 min (third left panel), 30 min (last panel). The nuclei fluorescence signal of DAPI is shown in the upper panel in blue, fluorescence of crosslinked proteins after click reaction to Alexa 555 (red) in the middle panel and a merge of both channels on the bottom. The images were also taken on an Olympus Spinning Disk Confocal microscope (2-024) using a magnification of 40 and a numerical aperture of 0.75.

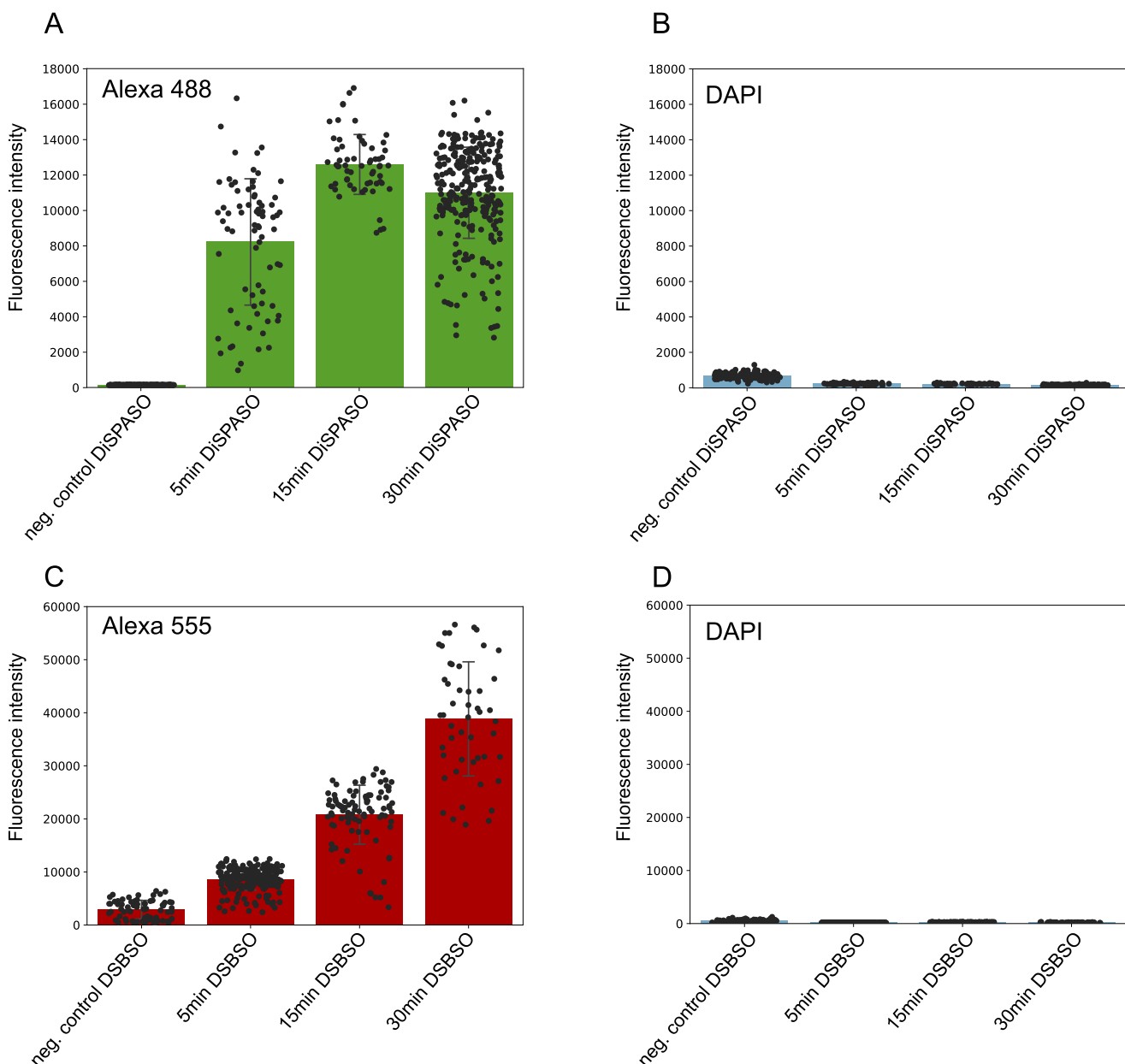

**Fig. 8 | Quantitation of crosslink fluorescence signals within nuclei of DiSPASO vs. DSBSO microscopy images. A** Quantitation of the green Alexa 488 signal of DiSPASO crosslinked proteins. The fluorescence intensity increases fast within the first 5 min after reaching a plateau at 30 min. Each dot represents a signal intensity of crosslinked proteins in a nucleus. **B** Nuclei control signal intensity of DAPI. The background intensity of DAPI is low in comparison to the intensity of crosslinked proteins. **C** Quantitation of the red Alexa 555 signal of DSBSO crosslinked proteins.

The fluorescence intensity increases slowly and reaches its maximum at 30 min. **D** Nuclei control signal intensity of DAPI. The background intensity of DAPI is also low in comparison to the intensity of crosslinked proteins. Quantitation of signal intensities of all images was performed in Fiji ImageJ (version 1.54f). The number of quantified nuclei is indicated as separate black dots on top of the bar. The standard deviation was estimated as average distance from each data point to the sample mean.

## Discussion

The introduction of DiSPASO in crosslinking mass spectrometry (XL-MS) has opened new avenues for elucidating protein-protein interactions (PPIs) within cells. DiSPASO has shown promising results via confocal microscopy, particularly with its successful cellular uptake and quick reactivity once inside the cell, highlighting its potential for in-depth cellular studies as well as revealing hurdles during mass spectrometry analysis.

In general, cleavable crosslinkers have labile bonds that break at lower energies than the peptide backbone, producing a distinct fragmentation pattern useful for analyzing complex mixtures. However, an excess of cleavage sites can reduce the identification rate, as observed with DiSPASO in our study. Consequently, one of the two signal doublets resulting from crosslinker cleavage may be absent, possibly due to further fragmentation or decreased signal intensity caused by an excess of fragment species in the spectra[20,46,48]. In contrast to the anticipated cleavage pattern, our findings reveal that DiSPASO exhibits unexpected additional cleavage sites beyond those expected for DSBSO, resulting in the formation of extra fragments. While it has been documented that carbon-sulfur bonds in benzyl mercaptans can rapidly dissociate under specific conditions, such as UVPD irradiation at 213 nm and 266 nm[49,50], this phenomenon was previously limited to certain wavelengths. Interestingly, recent studies have shown that C-S bond-selective photodissociation with 213 nm is augmented when sulfur is absent from an aromatic system by one methylene group ($sp^3$ carbon) but hampered when sulfur is directly attached to a $sp^2$ carbon. Surprisingly, our experiments indicate that this cleavage mechanism occurs even during standard HCD fragmentation.

Despite the challenges of complex fragmentation patterns and the need for refining data analysis, these findings offer valuable insights that drive further advancements in the field. The development of DiSPASO marks a significant step forward in enhancing cellular PPI mapping with high specificity and efficiency. However, the challenges encountered in translating its theoretical advantages into practical utility reveal a critical discrepancy that underscores the need for ongoing refinement in crosslinker technology. DiSPASO highlights the intricate balance between chemical innovation and biological functionality, emphasizing that advancements in crosslinker design must address current limitations to fully realize the potential of XL-MS in studying cellular mechanisms. While DiSPASO represents progress in PPI analysis, its limitations necessitate a cautious approach and highlight the complexity of developing effective crosslinking tools. Continuous innovation in structural design, particularly in simplifying crosslinkers and reducing potential cleavage sites, is essential. Additionally, advancements in computational tools and crosslinking search algorithms are crucial to overcoming challenges related to the search space in crosslinking data, enhancing the performance of non-cleavable crosslinkers for in-cell studies. These improvements will simplify data analysis and pave the way for more effective, streamlined XL-MS applications, ultimately bringing us closer to accurate and efficient mapping of PPI networks. Further studies involving different enrichment handles and reactive sites are ongoing to expand the capabilities of DiSPASO and similar crosslinkers.

## Methods

A list of all special reagents used within this study is given in Supplementary Table S1.

### Peptide synthesis

The Peptide Ac-WGGGGR**K**SSAAR-COOH was synthesized on a Liberty Blue peptide synthesizer (CEM) using standard Fmoc chemistry. For each amino acid cycle, a 4-min coupling with DIC/Oxyma was performed. N-terminal acetylation was performed by treating the final peptide on resin with 5% Acetic anhydride/2.5% DIPEA in DMF for 20 min. Peptide was purified on a Phenomenex Luna C18(2) using a 2–45% in 45 min 0.1%TFA/ACN+0.1%TFA gradient. The identity of the peptide was confirmed using MALDI-MS (4800 MALDI TOF/TOF, Sciex).

### Crosslinking reaction for Cas9

Cas9-Halo protein was crosslinked using either DSBSO or DiSPASO. All crosslinkers were prepared as stock solutions at a concentration of 20 mM in dry Dimethyl Sulfoxide (DMSO). For the crosslinking reaction, Cas9 was diluted in 50 mM HEPES to achieve a final protein concentration of 1 ug/uL and a crosslinker was added to a final concentration of 0.5 mM. After a 45-min incubation at room temperature, the reactions were stopped using 100 mM Tris buffer. Each crosslink reaction was prepared in parallel with the same conditions and buffers.

### Single peptide crosslinking

The peptide Ac-WGGGGR**K**SSAAR-COOH was dissolved in 50 mM HEPES buffer pH 7.3 to a final concentration of 5 mM and crosslinked using 2 mM DiSPASO. The crosslinker was added additionally 5x 2 mM over a time course of 2.5 h. The reaction was finally quenched with 100 mM Tris-buffer pH 8. Click reaction was performed using increasing amounts of picolyl azide: 2, 5, 10, 30 mM, 2 mM $CuSO_4$, 10 mM Tris[[1-Benzyl-1H-1,2,3-Triazol-4-yl)methyl]amin (TBTA) (later replaced by BTTAA for better solubility) and 100 mM Sodium Ascorbate (NaAsc). The procedure is described in more detail at point "Click reaction using Azide-S-S-biotin".

### In-solution digest

For the in-solution digest, ProteaseMax was utilized as a buffer component, prepared as a 1% stock solution in 50 mM Ammonium Bicarbonate (ABC) with a final concentration of 0.05%. Proteins were reduced using 10 mM Dithiothreitol (DTT) followed by incubation for 30 min at 50 °C, alkylated using 50 mM Iodoacetamide (IAA) for 30 min in the dark and finally digested using Trypsin (1:100, enzyme to protein ratio). The mixture was incubated overnight at 37 °C to facilitate complete digestion. The digestion process was terminated by the addition of 10% Trifluoroacetic Acid (TFA), adjusting to a final concentration of 0.5%, to ensure the degradation of ProteaseMax.

### Click reaction using Azide-S-S-biotin

For the click reaction, reagents were prepared from the CuAAC Kit provided by Jena Bioscience, including a 50 mM BTTAA stock solution, 1M stock of sodium ascorbate (NaAsc), 100 mM $CuSO_4$ stock solution in $H_2O$. A 50 mM Azide-S-S-biotin (BroadPharm) stock solution was prepared in dry DMSO. The click reaction utilized crosslinked Cas9 with DiSPASO at a concentration of 1 ug/uL, with the reaction environment maintained at pH 7.3. Initially, 2 mM $CuSO_4$ and 10 mM BTTAA were mixed, resulting in a blue solution. The Azide-S-S-biotin was then carefully added directly to the crosslinked Cas9 sample. The right amount of Azide was titrated to find an optimal azide concentration of 10 mM (Supplementary Fig. S3A). Following a short 5-min incubation period of azide and sample, the activated Cu(I) solution was added. NaAsc was then introduced to initiate the click reaction. The optimal NaAsc concentration was titrated as well with a resulting optimal concentration of 30 mM (Supplementary Fig. S3). The total reaction volume was adjusted with 50 mM HEPES pH 7.3 and incubated for 1 h at room temperature on a Thermo mixer set at 900 rpm.

### Crosslinked peptide enrichment

For Azide-S-S-biotin (ASSB) enrichment, the click reaction was performed in HEPES buffer at pH 7.3, using a mixture of 2 mM $CuSO_4$, 10 mM BTTAA, 10 mM ASSB, and 30 mM NaAsc. The reaction mixture was incubated for 1 h in the dark at room temperature. It was crucial to keep the temperature controlled, NaAsc concentration low, and the reaction shielded from light. The peptides were cleaned and desalted before the enrichment using self-made StageTips according to the procedure published before[51-53] and concentrated in an Eppendorf SpeedVac. For the bead enrichment process, 50 μL of MBS streptavidin beads, with a binding capacity of 1000 pmol free biotin per mg and a concentration of 5 mg/mL, were prepared at a bead-to-volume ratio of 1:5. The beads were washed three times with PBS. The sample, solubilized in PBS, was then added to the beads, and incubated for 1 h at room temperature, with the supernatant retained for binding

checks. Following this, the beads were washed three to four times with PBS to remove non-specific binders. The beads were then resuspended in 20 µL of PBS and eluted with 10 mM TCEP, followed by a 30-min incubation at room temperature. A second elution was performed with 20 µL of PBS and 10 mM TCEP. The eluted sample was then alkylated with IAA for 45 min in the dark at room temperature, followed by another desalting and cleanup step before mass spectrometry analysis.

Additionally, to the MBS beads, Pierce™ High-Capacity Streptavidin Agarose beads with 10 mg of biotinylated protein/mL and Pierce™ Streptavidin Magnetic Beads with ~55 ug biotinylated rabbit IgG/mg of beads or ~3500 pmol biotinylated fluorescein/mg of beads binding capacity were tested (Supplementary Fig. S4A, B).

### Ribosome crosslinked with DiSPASO

Enriched *E. coli* Ribosome obtained from BioLabs (S P0763S) at a concentration of 13.3 µM, was diluted to 1 mg/mL in 50 mM HEPES pH 7.3, 50 mM KCl, and 10 mM MgAc2. For the crosslinking procedure, 500 µL (500 µg) of this ribosome solution was treated with 1 mM DiSPASO, with the DiSPASO stock being freshly prepared at 30 mM in dry DMSO. The mixture was incubated for 30 min on ice, after which an additional 1 mM DiSPASO was added followed by another 30 min of incubation at room temperature. The reaction was then quenched with 1 M Tris to a final concentration of 100 mM Tris and left for 10 min. The sample was digested in solution with the addition of 0.05% ProteaseMax to the ribosome solution, followed by mixing and a 10-min incubation. The reduction of the ribosome solution was carried out using 10 mM DTT followed by 30 min incubation at 50 °C. The sample underwent water bath sonication for 3 min post-DTT reduction. Iodoacetamide (IAA) was added to a final concentration of 50 mM and incubated for 30 min in the dark. LysC was added at a ratio of 1:100 and incubated for 2 h at 37 °C, followed by the addition of trypsin at a ratio of 1:100 and overnight incubation at 37 °C. For desalting and cleanup, Sep Pak (50 mg, Waters) columns were used to desalt and remove excess crosslinker. The columns were activated with methanol (MeOH), washed, and equilibrated with 0.1% TFA. The sample (pH 3) was then loaded onto the column, washed twice with 0.1% TFA, and eluted with 80% ACN in 0.1% TFA. The ACN was subsequently evaporated using an Eppendorf SpeedVac. For the click reaction, the total volume was adjusted to 500 µL at pH 7.3, with a protein concentration of 1 µg/µL. Two mM CuSO$_4$ and 10 mM BTTAA were premixed as described above. Ten mM ASSB was added to the clean ribosomal peptides and click reaction initialization was performed by adding 30 mM NaAsc. The sensitivity experiment involved varying concentrations of ribosome and HEK in different ratios, namely 10:100, 2:100, 1:100, 0.5:100, and 0.25:100, with the control sample 10:100 (no enrichment). The ribosome quantities used ranged from 0.25 µg to 10 µg, while the HEK concentration was consistently maintained at 100 µg. Only the 1:100 ratio is shown in Supplementary Fig. S7 to exemplify that the enrichment did not work. Other rations have shown similar results.

### Sensitivity experiment using picolyl azide as click reagent

Cas9-Halo was crosslinked (200 µg protein) using 0.25 mM DiSPASO at a total protein concentration of 1 ug/uL. The reaction was carried out for 1 h at room temperature and quenched with 100 mM Tris buffer pH 8. Directly following the crosslinking, four volumes of pre-cooled acetone were added to the mixture to precipitate the protein. After incubation at −20 °C for 1 h, the mixture was centrifuged for 10 min at 14,000 × *g*. The supernatant was carefully removed without disturbing the protein pellet, which was then left to air dry at room temperature for 30 min to evaporate the remaining acetone. Over-drying was avoided to ensure the pellet's solubility. For in-solution digestion, the lysis buffer consisted of 8M Urea in 50 mM ABC. To this, 0.05% ProteaseMax was added, and the sample was vortexed and incubated for 10 min. The protein sample was reduced using 10 mM Dithiothreitol (DTT), for 30 min at 50 °C following alkylation utilizing 50 mM Iodoacetamide (IAA) for 30 min in the dark. The Urea concentration was reduced to 2 M before protein digestion. Proteolysis was initiated with LysC at a ratio of 1:100 (LysC:protein) for 2 h at 37 °C, followed by Trypsin at a 1:100 ratio and overnight incubation. For desalting and free crosslinker removal, Sep Pak

(50 mg) columns were utilized. The columns were activated with Methanol and equilibrated with 0.1% TFA. After loading the sample, it was washed twice with 0.1% TFA and eluted with 80% ACN in 0.1% TFA. The clean peptides were dried using an Eppendorf SpeedVac. HeLa Lysate, used as a background, was digested and cleaned in the same way.

The sensitivity experiment mixtures contained a constant background of 1 mg HeLa digest with increasing spike-in of crosslinked Cas9 samples. The following ratios have been used to determine the sensitivity: 10:100 (1:10), 5:100, 1:100 and 0.5:100. The click reaction setup involved a total volume of 500 µL at pH 7.3, with a total protein concentration of 2 µg/µL. The reaction contained 2 mM CuSO4, 10 mM BTTAA, 5 mM picolyl azide and 100 mM Sodium Ascorbate and was incubated for 1 h at room temperature. After clean-up of the clicked and crosslinked peptides an enrichment procedure was performed, involving the use of TiO$_2$-beads.

For the enrichment process, the beads-to-peptide ratio was maintained between 8:1 and 10:1, with a minimum of 10 mg of beads utilized for each condition. Offline columns were assembled with a 10 µm filter and beads weighed directly into them. These beads were first suspended in 200 µl of 50% methanol and centrifuged using a tabletop centrifuge for 1 min at 2000 × *g*. The beads were then resuspended in 200 µl water, and centrifuged again, followed by two rounds of resuspension in 200 µl 1M glycolic acid solution (1M glycolic acid in a mixture of 70% ACN and 3% TFA) and centrifugation. The beads were finally resuspended in 200 µl of the glycolic acid solution. Concentrated and cleaned samples (200 uL final volume), were mixed with an equal volume of the glycolic acid solution and transferred into 1.5 ml tubes. The bead slurry was added to reach a total volume of 700 µl, vortexed, and incubated on a Thermo-Mixer at room temperature for 30 min at 1000 rpm. Following incubation, the samples were centrifuged in the offline column. The washing process involved resuspending the beads in various solutions and centrifuging them between each step. Initially, the beads were washed twice with 200 µl glycolic acid solution, followed by two washes with 200 µl 70% ACN/3% TFA, and then two washes with 200 µl 1% ACN/0.1% TFA. For elution, the beads were resuspended in 150 µl 300 mM NH$_4$OH and incubated for 1 min before centrifugation. The elution step was repeated twice. The eluate was neutralized with 5–7.5 µl concentrated TFA.

### In-cell crosslinking of HEK and HeLa cells using DiSPASO

In the conducted experiment, the cell culture was established using Dulbecco's Modified Eagle Medium (DMEM) supplemented with 10% Fetal Bovine Serum (FBS) (50 mL) (10270, Fisher Scientific, USA), 1% Penicillin/Streptomycin (5 mL) (P0781-100ML, Sigma-Aldrich, Israel), and 2 mM (1%) L-Glutamine (5 mL) (250030-024, Thermo Scientific, Germany). The frozen HEK 293 cells were taken in culture in 6-well plates. Key steps included centrifugation of the cell suspension and resuspension of the cell pellet in fresh media. After cell culture for at least 2 days (around 80–90% confluency) in the incubator at 37C and 5% CO$_2$, the cells were split, involving the removal of old media, washing with phosphate-buffered-saline (PBS), addition of 0.05% Trypsin-EDTA (25300-054, Thermo Scientific, USA) solution for digestion of surface proteins, preparation of new dishes with new DMEM medium, and addition of cell suspension to new dishes. For in-cell crosslinking, the cells were washed 2x times with PBS and the crosslinker (DiSPASO) was added directly to the cells in each dish. To keep the DMSO concentration below 5%, the crosslinker stock solution was diluted in PBS to a final concentration of 5 mM and immediately added to the cells after washing. Keeping the DMSO concentration low ensures that the cells stay intact and avoids crosslinking of broken or damaged cells. After incubation for 30 min in an incubator the reaction was quenched with 100 mM Tris-buffer for 5 min. Afterwards, the cells were washed again with PBS detached with the addition of Trypsin solution and incubated for several minutes at 37 C and 5% CO$_2$. The cells were collected in a new reaction tube, washed again with PBS and centrifuged at 300 × *g* for 2 min. Cells were resuspended in a lysis buffer (50 HEPES pH 7.3, 8M Urea and 1% Dodecyl maltoside). After 3 times sonication for 30 s (amplitude 80%, 0.5 s cycle, UP100H Ultrasonic Processor, Hielscher), the samples were subjected to reduction using Dithiothreitol (DTT) with a final concentration of 10 mM, followed by an incubation period of 30 min at 50 °C.

Subsequently, Iodoacetamide (IAA) was added to reach a final concentration of 50 mM, and the sample was then incubated for 30 min at room temperature in the dark. Next, the sample was diluted to a concentration of 2M Urea using 50 mM HEPES pH 7.3. LysC enzyme was added at a ratio of 1:100, and the mixture was incubated at 37 °C for 2 h. Following, Trypsin digest at a ratio of 1:100, and incubation at 37 °C overnight. After the digestion, the sample was desalted and cleaned as described earlier.

## Crosslink enrichment using Disulfide Azide Agarose beads (DAAB)

For the DAAB enrichment, clean crosslinked peptides were diluted to 2 ug/uL using HEPES buffer pH 7.3. The click reaction was carried out directly on the beads. Disulfide beads were then added to the sample, followed by a brief incubation period of the beads and sample. The activated Cu(I) solution was added to the sample (2 mM $CuSO_4$ and 10 mM BTTAA), and 30 mM sodium ascorbate (NaAsc) was introduced to initiate the click reaction. The sample was then incubated for 2 h at room temperature on a rotation wheel to ensure thorough mixing of the beads. Following the incubation period, the beads were washed with PBS, repeating this step three times. Elution was carried out using 10 mM TCEP (final concentration), with an incubation period of 1 h at room temperature, followed by the transfer of the supernatant to a new tube. This elution step was repeated with fresh TCEP, incubating again for 1 h at room temperature, and the elution was subsequently combined. To facilitate the alkylation of the free disulfide group, 50 mM IAA (final concentration) was added to the elution, and the sample was incubated for 45 min at room temperature in the dark.

## Sample preparation for confocal microscopy of crosslinked HEK cells

The labeling and crosslinking of HEK cells proceeded as follows: Old media was removed, and cells were washed once with PBS. Five mM DiSPASO in PBS was added and incubated for a specified time (0 min (control), 5 min, 15 min, 30 min). After each time point, cells were washed with 1 mL PBS and incubated with 200 uL 100 mM Tris in PBS for 10 min. Subsequently, cells were washed twice with 1 mL PBS and fixed with 1 mL 3.7% formaldehyde solution in PBS for 15 min at room temperature. Following fixation, cells were washed twice with 1 mL 3% BSA in PBS, and then 1 mL PBS-T (0.5% Triton X-100 in PBS) was added, with an incubation period of 20 min at room temperature. For the click reaction using the Click-iT Alexa Fluor Picolyl Azide Toolkit, cells were washed twice with PBS. The Alexa Fluor 488 PCA stock solution (950 nominal molecular weight) in 210 uL DMSO was prepared to achieve a final concentration of 500 uM. A mixture of 2 uL of $CuSO_4$ (Compound C) with 8 uL Copper protectant (Compound D) was prepared for a single click reaction. The total volume of the click reaction was adjusted to 500 uL, comprising 435 uL of 1x click reaction buffer (Compound B), 5 uL of 500 uM Alexa Fluor (final 5 uM), 10 uL $CuSO_4$-Copper protectant pre-mix, and 50 uL 1x click buffer additive (Compound E), resulting in a final DMSO concentration from the Alexa Fluor of 1%. For wells containing DSBSO crosslinked HEK 293 cells, AlexaFluor 555 alkyne was added instead of AlexaFluor 488. Cells were then incubated for 2 h at 37 °C in the dark. Post-incubation, cells were washed twice with PBS, and DAPI was added for 2 min before washing twice with PBS again. Finally, cells were stored in PBS at 4 °C until microscopy, with precautions taken to prevent fluorophore bleaching.

## Confocal microscopy procedure

Images were recorded using a spinning disc confocal scan head (Yokogawa, W1) mounted on Olympus IX3 microscope (Olympus). Multicolor images were acquired using the orca flash 4 camera (Hamamatsu). 40x/0.75 UPLFN (Olympus) was used for images in Fig. 7, 7/1.2W UPLSAPO (Olympus) and 100x/1.45O UPLXAPO (Olympus) were used for illustrations in Supplementary Fig. S5 left and right respectively. DAPI, AlexaFluor 488 and AlexaFluor 555 were excited using 405 nm, 488 nm and 561 nm lasers respectively. All images were acquired with the same imaging conditions within each experimental group. The exposure times for fluorescence

measurements were set to 400 ms for DSBSO (405 nm, 561 nm) and 100 ms (405 nm) and 50 ms (561 nm) for DiSPASO.

## Relative quantitation of fluorescence signals

For quantifying the fluorescent intensities, we used a custom Fiji-macro[54]. Measurements were performed in 2D, using the central optical slice of each acquisition. Nuclear segmentation was done via the StarDist-plugin (https://github.com/stardist/stardist). Border objects and small fragments were excluded. Intensities for both channels were then measured within the segmented regions.

## Mass spectrometry

LC-MS/MS analysis was performed using an Orbitrap Exploris 480 or Orbitrap Eclipse Tribrid mass spectrometer with Field asymmetric ion mobility spectrometry (FAIMS) interface (Thermo Fisher Scientific, Waltham, Massachusetts, United States) coupled with a Dionex UltiMate 3000 HPLC system (Thermo Fisher Scientific, Waltham, Massachusetts, United States). A trap column PepMap C18 (5 mm × 300 μm ID, 5 μm particles, 100 Å pore size) (Thermo Fisher Scientific, Waltham, Massachusetts, United States) and an analytical column PepMap C18 (500 mm × 75 μm ID, 2 μm, 100 Å) (Thermo Fisher Scientific, Waltham, Massachusetts, United States) were employed for separation. The column temperature was set to 50 °C. Sample loading was performed using 0.1% trifluoroacetic acid in water with a flow rate of 50 uL/min for 3 min. Mobile phases used for separation were as follows: (A) 0.1% formic acid (FA) in water; (B) 80% acetonitrile, 0.1% FA in water. Peptides were eluted using a flow rate of 230 nL/min, with the following gradient: from 2% to 45% phase B in 90 min, from 45% to 95% phase B in 1 min, followed by a washing step at 95% for 6 min, and re-equilibration of the column. The gradient was altered over time for optimization purposes. The gradient stated here was the best performing in our hands.

FAIMS separation was performed with the following settings: inner and outer electrode temperatures were 100 °C, FAIMS carrier gas flow was 4.6 L/min, compensation voltages (CVs) of −50, −60, and −70 V were used in a stepwise mode during the analysis. The mass spectrometer was operated in a data-dependent mode with cycle time 2 s, using the following full scan parameters: $m/z$ range 375–1500, nominal resolution of 120,000, with a target of 250% charges for the automated gain control (AGC), and automated selection of maximum injection time. For higher-energy collision-induced dissociation (HCD) MS/MS scans, a stepped normalized collision energy (NCE) of (25%; 27%; 32%) for DiSPASO and 21%; 27%; 32%) for DSBSO, the resolution was 30,000. Precursor ions were isolated in a 1.4 Th window with no offset and accumulated for a maximum of 54 ms or until the AGC target of 100% was reached. Precursors of charge states from 3+ to 6+ were scheduled for fragmentation. Previously targeted precursors were dynamically excluded from fragmentation for 25 s. The sample load was typically 500 ng and 200 ng. Detailed parameters can be found in each raw file under the instrument method section.

## Data analysis

Raw files were analyzed using Thermo Proteome Discoverer (v. 3.1.0.638). Searches were performed against the Cas9 sequence (Uniprot ID: Q99ZW2) plus a Crapome database (downloaded from https://www.thegpm.org/crap/). For in-cell crosslinking searches, fasta files were created for each experiment series separately by searching against the full Human database (Uniprot ID UP000005640, last update 04.08.2022, 20528 sequences). Identified proteins with less than 3 peptide spectrum matches (PSM) were filtered out. For the ribosome samples, the *E. coli* K12 database (Uniprot IDUP000000625, last update 24.04.2023, 5296 sequences) was used to create a specific database for crosslink searches.

Linear peptides were identified using MS Amanda search engine (v. 3.0.20.558)[55–57] and the crosslinked peptides were identified using MS Annika (v. 2.2.2)[58,59]. The search workflow included a recalibration step for each file, followed by a first search using MS Amanda to identify linear peptides and monolinks. Subsequently, spectra with highly confident

identifications of a linear peptide were filtered out and not considered for the cross-link search. Finally, a crosslink search was performed using MS Annika. The workflow used in Proteome Discoverer is shown in Supplementary Fig. S6. Search parameters for linear and crosslink searches can be found in Supplementary Tables 2 and 3. The FDR was estimated using the MS Annika validator node with 1% FDR (high confidence) for all single peptide and Cas9 crosslinking data and 5% (high and medium confidence) for all in-cell crosslinking data on CSM and residue pair levels. The FDR calculation is based on a target-decoy approach[59].

For data filtering and visualization Python 3.9.7 was used with the following packages: pandas[60], numpy[61], matplotlib[62] (pyplot, venn), seaborn[63], scipy and bioinfokit[64,65].

### Software adjustments
We have updated MS Annika, our cross-linking search engine, to better accommodate the fragmentation behavior of DiSPASO. By adjusting the "Additional Crosslink Doublet Distances" parameter, multiple doublets can now be searched in a single run, provided they share the same light fragment as specified in the crosslinker definition in Proteome Discoverer. For instance, in one search run, it is feasible to explore doublets such as alkene–thiol, alkene–sulfenic acid, alkene–ETFP, alkene–ETHMP, and alkene–full crosslinker by designating alkene–thiol as the default doublet and adding the remaining doublet distances in the MS Annika settings. However, including a smaller doublet in the same run is not feasible as it would alter the role of the alkene fragment due to its lower mass. The additional doublet distances are determined by subtracting the monoisotopic mass of the lighter fragment from that of the heavier one. Additionally, MS Annika now considers all potential crosslinker fragments as peptide modifications during search, further enhancing crosslink identification. This entails specifying all expected fragments as neutral losses in the crosslinker definition in Proteome Discoverer, allowing MS Annika to calculate corresponding theoretical ions for precise identification during the database search.

Equations (1)–(4) show an exemplary calculation of the doublet distance for the doublet alkene–sulfenic acid (A–SA). The subtraction of the mass of the lighter fragment from the mass of the heavier fragment gives the doublet distance. For the case of the A–SA doublet, subtracting the mass of the alkene fragment from the mass of the sulfenic acid fragment yields a doublet distance of approximately 49.98 (Da).

$$Doublet\ Distance = Mass(fragment_{heavy}) - Mass(fragment_{light}) \quad (1)$$

$$Doublet\ Distance_{A-SA} = Mass(sulfenic\ acid) - Mass(alkene) \quad (2)$$

$$Doublet\ Distance_{A-SA} = 119.0041 - 59.02146 \quad (3)$$

$$Doublet\ Distance_{A-SA} = 49.98264 \quad (4)$$

### Surface area plot creation
To determine the lipophilicity and other physicochemical properties of crosslinker compounds, a Python script utilizing RDKit and pandas libraries was used. First, the cLogP (partition coefficient) of the compounds was calculated. The cLogP value quantifies a compound's tendency to partition between an organic solvent (typically octanol) and water, reflecting its lipophilicity. This was achieved by defining a function *calculate_clogp* that utilizes RDKit's *MolFromSmiles* and *MolLogP* functions to calculate the cLogP value for each compound represented by its SMILES notation. Similarly, the topological polar surface area (tPSA) was calculated using the function *calculate_tpsa*, which utilizes RDKit's *CalcTPSA* function to compute the tPSA based on the molecule's topology. Additionally, a scatter plot was generated to visualize the relationship between cLogP and tPSA values for all crosslinker compounds. The plot provides insights into the correlation between the lipophilicity and polar surface area of the compounds, which are crucial factors in drug design and membrane permeability prediction. Furthermore, the partition coefficient (*P*) highlights its significance in understanding a compound's hydrophobicity.

### Reporting summary
Further information on research design is available in the Nature Portfolio Reporting Summary linked to this article.

### Data availability
The mass spectrometry proteomics data and raw microscopy images have been deposited to the ProteomeXchange Consortium (http://proteomecentral.proteomexchange.org) via the PRIDE partner repository[66] with the dataset identifier PXD056091. NMR spectra of all synthesized compounds are reported in the Supplementary Data 1.zip file. Tables 1–3 summarize the unique residue pairs identified across all experiments and data that have been used to plot the figures in this manuscript.

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

## Acknowledgements

This work was supported by the infrastructure funding 4th call 2022/01 (AT-SCP) of the Austrian Research Promotion Agency (FFG) and the project LS20-079 of the Vienna Science and Technology Fund (WWTF). This work was further funded by the ESPRIT program project number ESP 566 (Grant-DOI 10.55776/ESP566), P35045-B project (Grant-DOI 10.55776/P35045) and the F 8801-B Meiosis project (Grant-DOI 10.55776/F88) of the Austrian Science Fund (FWF). All LC-MS/MS analyses in Vienna were performed on the Vienna BioCenter Core Facilities instrument pool. We thank the Vienna Biocentre BioOptics facility for help and advice with microscopy imaging. Synthesis was performed at the Institute of Organic Chemistry of the University of Vienna. Funding from the Austrian Academy of Sciences (DOC Fellowship to B.R.B.) is acknowledged. We thank the University of Vienna for its generous support. This research was funded in whole, or in part, by the Austrian Science Fund (FWF). For the purpose of open access, the author has applied a CC BY public copyright licence to any Author Accepted Manuscript version arising from this submission.

## Author contributions

Fränze Müller conceptualized the study, designed the MS and microscopy-based experiments, performed data analysis, wrote the manuscript (MS based & microscopy based), designed the figures; Bogdan R. Brutiu synthesized the DiSPASO cross-linker and wrote the chemistry part of the manuscript; Iakovos Saridakis wrote the chemistry part of the FWF proposal and designed the cross-linker; Thomas Leischner synthesized the picolyl azide; Micha J. Birklbauer adapted MS Annika for additional doublet searches and wrote the corresponding method section; Manuel Matzinger conceptualized and wrote the main part of the FWF proposal and provided data for DSBSO analysis; Mathias Madalinski synthesized the peptide used in this study and wrote the corresponding method section; Thomas Lendl quantified the fluorescence signals of microscopy experiments and wrote the corresponding method section; Saad Shaaban coordinated the chemical experiments and wrote the chemistry part of the manuscript; Karl Mechtler, Nuno Maulide and Viktoria Dorfer supervised the study. All authors revised and agreed on the manuscript.

## Competing interests

The authors declare no competing interests.
