## [Transparent Peer Review file · Communications Chemistry]

Developing a new cleavable crosslinker reagent for in-cell crosslinking

Corresponding Author: Professor Karl Mechtler

Version 0:

Reviewer comments:

Reviewer #1

(Remarks to the Author)

The manuscript by Müller et al. discusses the development of a new cleavable crosslinking reagent, DiSPASO, for in-cell crosslinking and the challenges faced.

DiSPASO was designed with N-hydroxysuccinimide groups as reactive sites and an alkyne group in the spacer for enrichment purposes after click chemistry coupling. Sulfoxide groups serve as cleavage sites. Therefore, several building blocks that have already been successfully used in crosslinker design were combined and the properties of DiSPASO promised cell membrane permeability, which was also confirmed by confocal microscopy. However, practical applicability to detect large numbers of protein-protein interactions in cells proved difficult. Different enrichment strategies were evaluated, e.g. attaching a phosphonate handle for enrichment by TiO₂ affinity chromatography, attaching a chemically cleavable biotin affinity tag or direct coupling to azide beads via a disulfide group-containing linker. However, the different approaches suffered from limited enrichment efficiencies, and the mass spectrometric dissociation pattern of the linker proved to be complex, resulting in reduced identification rates.

In summary, this is an interesting report discussing the design, synthesis and application of a new crosslinking reagent for in-cell applications. The study also introduces different enrichment strategies based on click chemistry, which may also be applied to other reagent designs. The relatively poor practical performance highlights the challenges of developing new reagents as the real-life applicability remains to some extent unpredictable. I consider the manuscript suitable for publication in Communications Chemistry. It is well written and provides a wealth of information in the main text and the SI. I do not have any major concerns, but a number of minor issues should be addressed in a revised version.

Specific comments:

Abstract: What are "harsh fragmentation operations"? Please explain/revise. The final statement of the abstract, referring to "requiring a careful strategy", does not really capture the outcome of the study - even with careful optimization, the number of crosslinks remained low.

Page 6: The selection of collision energies is not logical. The highest score was obtained for a CE of 34, yet stepped CE of 25/27/32 was used for further experiments?

Page 7, legend to Figure 4: What is "high-performance CID"? Do the authors mean HCD, i.e. higher-energy CID?

Page 9: "The strong binding of the biotin group ..." This sentence is unclear, please revise.

Page 10: The ascorbate concentration was optimized to "minimize" side reactions. Which ones? How were they assessed?

Page 13, legend to Figure 8: When referring to crosslinked PEPTIDES, probably the authors mean crosslinked PROTEINS here? While in A, one can argue that "a plateau" is reached, this is definitely not the case for C?

Page 17, Ref. 16: The journal name is eLife, not eLife Sciences.

Page 18, Ref. 21: Article details such as journal, volume etc. are missing.

SI pages 25-26: References to "SubFig.X" need to be updated.

SI page 27: Please avoid the term "stage tipping" when referring to other designs such as columns or cartridges. The procedure is called solid-phase extraction.

SI page 28: What are "mobile columns"? Please explain. Regarding the titanium dioxide enrichment, did you also consider dephosphorylating endogenous phosphopeptides to reduce co-enrichment?

SI page 29: Please provide the type of sonication device used.

SI page 34, Figure S3: In panel A, 10 mM ASSB is not shown although mentioned in the legend. In C, the 30 mM selected as optimal are beyond the range evaluated and shown here. In F, the value for 50 uL of beads is not shown, although it is mentioned in the legend.

Typos:

Page 5, legend to Figure 2: shoes > shows

Page 9: electron spray > electrospray

SI page 25: Thermo Fischer > Thermo Fisher

SI page 30: TECEP > TCEP

SI page 34: sodium adsorbate > sodium ascorbate

Reviewer #2

(Remarks to the Author)

The author proposed an interesting cross-linking reagents for XL-MS. Overall the cross-linker showed good trans-membrane and enrichment properties with sufficient evidence. However, apart from minor issues, my main concern with this study is the the lack of comprehensive model compound (peptide) data, which should be address before I can recommend the manuscript for publication.

In the study, cross-linking on only one peptide is studies prior to HeLa cell XL-MS. I recommend the authors include more peptides and test the cross-linker on peptide mixtures without and with matrix (e.g. HeLa cell lysate) to better understand, evaluate, and optimize the click chemistry.

In addition, there are also some minor issues:

1. More proof reading on some simple phrases such as "MS-cleavable cleavable cross-linkers"
2. It will be better to also include mass spectra in the supplementary data.

Reviewer #3

(Remarks to the Author)

The authors of this manuscript present a comprehensive study on the development and characterization of DiSPASO, a novel lysine-reactive, MS-cleavable crosslinker. The linker includes an affinity handle, and several strategies have been tested to maximize the number of identifications. The work includes testing the reagent at increasing levels of sample complexity, up to live cells, and a side-by-side comparison with a similar reagent that has already been published. The manuscript is well-structured, with detailed experimental evidence supporting both the crosslinker's effectiveness and its limitations. I appreciate the manuscript as it also discusses unsuccessful attempts, highlighting the complexity of developing a new crosslinker. I believe this manuscript makes a valuable contribution to the field of crosslinking mass spectrometry and can be accepted with the following minor revisions, provided the authors make the data publicly available (PXD056091 is not currently accessible, so I could not review the data).

Page 2: "MS-cleavable cleavable cross-linkers" should be revised to "MS-cleavable cross-linkers."

Page 4: "shows medium membrane permeability" should be revised to "shows moderate membrane permeability."

Page 5: "If a crosslinker shoes two properties" should be corrected to "If a crosslinker shows two properties."

Reviewer #4

(Remarks to the Author)

In this work, the authors developed new cleavable crosslinker reagents for in-cell protein crosslinking and evaluated their performance for XL-MS. As the authors also noted, the reagents developed have many challenges and do not offer any particularly noteworthy advantages compared to existing reagents. Although XL-MS reagents with novel molecular structures may provide insights into future development and optimization in XL-MS applications, the current study is too preliminary to be published in Communications Chemistry, where high academic impact and novelty are required.

Therefore, I recommend that this manuscript should be submitted to other journals more specific to the field of mass spectrometry.

Minor comments:

1. There are many typos in the manuscript. For example,
-MS-cleavable cleavable cross-linkers
-an alkyne-based click chemistry, not "alkene"
2. Figure 3: There is no nitrogen atom on the benzene ring of Picoryl-Azide (3).
3. Figure 3: The structure of the BAPD (8) eluted from the beads using a reducing agent is incorrect. It should not be cleaved at the amide bond.

Version 1:

Reviewer comments:

Reviewer #1

(Remarks to the Author)

The authors have addressed most of my comments appropriately in this revised version. I am still not convinced with the sentence in lines 285-288:

"The strong binding (free binding energy of -41.17 kcal/mol⁴⁷, K_d of ~ 10 – 30 M) of the biotin group to streptavidin beads can be fully exhausted, while binding efficiency can be granted due to the reduction of the disulfide bond."

The binding can be exhausted - exhausted means tired, maybe also depleted in this context. However, I think what the authors mean here is that the strong binding ensures quantitative capture of the compound(s) containing a biotin group.

Binding efficiency can be granted due to the reduction of the disulfide bond - what does binding EFFICIENCY have to do with reduction of the disulfide bond? Maybe the authors meant reversibility of the binding?

Reviewer #2

(Remarks to the Author)

The author presented an interesting strategy and honestly reported the process for developing a new reagent for cross-linking mass spectrometry. The author to some extent addressed my concern. However, it will be better to optimize model compounds with or without the context of cell lysate to pinpoint the reason for low cross-linking efficiency.

Nevertheless, this study is well-designed and indeed every MS cross-linking reagents inevitably requires extensive optimization and lengthy, continuous efforts. Sharing such knowledge and process is important for scientific communication. In this context, I can recommend this manuscript to be published.

Reviewer #3

(Remarks to the Author)

The authors have satisfactorily addressed all the points raised during the review process, thereby improving the manuscript. I believe it is now suitable for publication in Communications Chemistry.

A journey towards developing a new cleavable crosslinker reagent for in-cell crosslinking

We want to thank the reviewers for taking the time and effort to review our manuscript and provide detailed feedback.

Below you will find a point-by-point response to the comments made by the reviewers with our answer in green.

Reviewers' comments:

Reviewer #1 (Remarks to the Author):

The manuscript by Müller et al. discusses the development of a new cleavable crosslinking reagent, DiSPASO, for in-cell crosslinking and the challenges faced.

DiSPASO was designed with N-hydroxysuccinimide groups as reactive sites and an alkyne group in the spacer for enrichment purposes after click chemistry coupling. Sulfoxide groups serve as cleavage sites. Therefore, several building blocks that have already been successfully used in crosslinker design were combined and the properties of DiSPASO promised cell membrane permeability, which was also confirmed by confocal microscopy. However, practical applicability to detect large numbers of protein-protein interactions in cells proved difficult. Different enrichment strategies were evaluated, e.g. attaching a phosphonate handle for enrichment by TiO₂ affinity chromatography, attaching a chemically cleavable biotin affinity tag or direct coupling to azide beads via a disulfide group-containing linker. However, the different approaches suffered from limited enrichment efficiencies, and the mass spectrometric dissociation pattern of the linker proved to be complex, resulting in reduced identification rates.

In summary, this is an interesting report discussing the design, synthesis and application of a new crosslinking reagent for in-cell applications. The study also introduces different enrichment strategies based on click chemistry, which may also be applied to other reagent designs. The relatively poor practical performance highlights the challenges of developing new reagents as the real-life applicability remains to some extent unpredictable. I consider the manuscript suitable for publication in Communications Chemistry. It is well-written and provides a wealth of information in the main text and the SI. I do not have any major concerns, but a number of minor issues should be addressed in a revised version.

Specific comments:

Abstract: What are "harsh fragmentation operations"? Please explain/revise. The final statement of the abstract, referring to "requiring a careful strategy", does not really capture the outcome of the study - even with careful optimization, the number of crosslinks remained low.

The abstract was revised to: "including extensive evaluation of fragmentation energies and fragmentation behaviour of the crosslinker backbone." and "its limitations and low crosslinking yield in cellular environments require careful optimisation of the crosslinker design"

Page 6: The selection of collision energies is not logical. The highest score was obtained for a CE of 34, yet stepped CE of 25/27/32 was used for further experiments?

The collision energy shows a general decrease of doublet signal intensities towards higher energies (except for NCE 34), stepped CE was used to facilitate high doublet intensities while getting sufficient peptide backbone fragmentation. A CE of 32 is enough to get proper peptide backbone fragmentation, 25 and 27 have been used to maintain high intensities for the alkene-SA as well as the alkene-thiol doublet. We refer here to the paper of Steigenberger et al. 2019, who have extensively tested the dependence of doublet fragments on HCD/CID energies. CE 34 was not selected as the maximum value due to the increase in intensity of the doublet although the trend showed decreasing intensities with higher CE. Since CE 32 is enough to get good backbone fragmentation, we stuck to the lower value.

Kolbowski *at al.* have been testing the fragmentation behaviour of crosslinked peptides as well, and the optimal HCD fragmentation energy was determined as 30 on a Tribrid Lumos mass spectrometer. Even so, 32 is higher than the energy of 30, we believe that this small difference in fragmentation energy does not change the performance of DiSAPSO in our study.

References:

Barbara Steigenberger, Herbert B. Schiller, Roland J. Pieters, Richard A. Scheltema, Finding and using diagnostic ions in collision-induced crosslinked peptide fragmentation spectra, *International Journal of Mass Spectrometry*, Volume 444, 2019, 116184, ISSN 1387-3806, <https://doi.org/10.1016/j.ijms.2019.116184>.

Kolbowski L, Mendes ML, Rappsilber J. Optimizing the Parameters Governing the Fragmentation of Cross-Linked Peptides in a Tribrid Mass Spectrometer. *Anal Chem*. 2017 May 16;89(10):5311-5318. doi: 10.1021/acs.analchem.6b04935. Epub 2017 Apr 26. PMID: 28402676; PMCID: PMC5436099.

Page 7, legend to Figure 4: What is "high-performance CID"? Do the authors mean HCD, i.e. higher-energy CID?

We changed the text accordingly to fit the HCD description.

Page 9: "The strong binding of the biotin group ..." This sentence is unclear, please revise.

We have revised this sentence to: The strong binding (free binding energy of -41.17 kcal/mol(Liu et al. 2016), K_d of $\sim 10^{-30}$ M) of the biotin group to streptavidin beads can be fully exhausted, while binding efficiency can be granted due to the reduction of the disulfide bond.

References:

Liu, F., Zhang, J. & Mei, Y. The origin of the cooperativity in the streptavidin-biotin system: A computational investigation through molecular dynamics simulations. *Sci Rep* 6, 27190

(2016).

<https://doi.org/10.1038/srep27190>

Page 10: The ascorbate concentration was optimized to "minimize" side reactions. Which ones? How were they assessed?

In the literature it is reported that sodium ascorbate can cause side reactions like:

a) Copper ions mediate the oxidation of sodium ascorbate by molecular oxygen, producing hydrogen peroxide (H_2O_2) via a two-step process that involves the superoxide radical anion. This can lead to the oxidation of biomolecules, particularly peptides. Certain amino acid residues like cysteine, methionine, and histidine imidazole groups are especially susceptible to this oxidation.

b) Dehydroascorbate, an initial oxidation product of sodium ascorbate, is a potent electrophile. It can hydrolyse to form reactive aldehydes, such as 2,3-diketogulonate and glyoxal. These byproducts can react with the amine groups of lysine and the guanidine groups of arginine, which can cause covalent modifications, crosslinking, and potential aggregation of proteins.

References:

Hong V, Presolski SI, Ma C, Finn MG. Analysis and optimization of copper-catalyzed azide-alkyne cycloaddition for bioconjugation. *Angew Chem Int Ed Engl.* 2009;48(52):9879-83. doi: 10.1002/anie.200905087. PMID: 19943299; PMCID: PMC3410708.

Presolski SI, Hong VP, Finn MG. Copper-Catalyzed Azide-Alkyne Click Chemistry for Bioconjugation. *Curr Protoc Chem Biol.* 2011;3(4):153-162. doi: 10.1002/9780470559277.ch110148. Epub 2011 Dec 1. PMID: 22844652; PMCID: PMC3404492.

The H_2O_2 yield during click reaction can be minimised by using a ligand that not only stabilises Cu(I) but also can capture H_2O_2 . In our case, we have used BTAA. This ligand can act as a sacrificial reductant by intercepting reactive oxygen species in the coordination sphere of the metal. Using an excess of ligands is crucial for this protective effect, which we did. It also helps to protect biomolecules from oxidation. We additionally tried Aminoguanidine which can capture reactive carbonyl compounds, like the byproducts of ascorbate oxidation, thus preventing their unwanted reactions with biomolecules (here peptides), but this procedure did not improve the crosslinking yield after enrichment. Indeed, the efficiency of the click reaction was reduced.

We have checked the possibilities of byproducts/ modifications of amino acids using open modification searches of our sodium ascorbate optimization samples. The side reaction yield was in general low for sodium ascorbate concentrations of less than 100mM. The side reactions seem to not cause the low yield of unique crosslinks after enrichment, therefore we did not further investigate in this direction.

Page 13, legend to Figure 8: When referring to crosslinked PEPTIDES, probably the authors mean crosslinked PROTEINS here? While in A, one can argue that "a plateau" is reached, this is definitely not the case for C?

The legend was changed accordingly. Figure C indeed did not reach a plateau but its maximum at 30min. The text was changed accordingly.

Page 17, Ref. 16: The journal name is eLife, not eLife Sciences.

Reference journal names were changed.

Page 18, Ref. 21: Article details such as journal, volume etc. are missing.

Ref was updated

SI pages 25-26: References to "SubFig.X" need to be updated.

Changed accordingly

SI page 27: Please avoid the term "stage tipping" when referring to other designs such as columns or cartridges. The procedure is called solid-phase extraction.

Stage tipping was replaced by desalting and cleanup

SI page 28: What are "mobile columns"? Please explain. Regarding the titanium dioxide enrichment, did you also consider dephosphorylating endogenous phosphopeptides to reduce co-enrichment?

Mobile column has been changed to offline column. These columns are not installed in an HPLC system but rather used as separate device.

We did not consider dephosphorylation. Although dephosphorylation does improve the enrichment of crosslinked peptides by 22% (Jiang *et al.* 2022), this would not circumvent the hurdles that are introduced by the fragmentation behaviour of DiSPASO and therefore also not dramatically increase the crosslink yield after enrichment.

Reference:

Jiang PL, Wang C, Diehl A, Viner R, Etienne C, Nandhikonda P, Foster L, Bomgarden RD, Liu F. A Membrane-Permeable and Immobilized Metal Affinity Chromatography (IMAC) Enrichable Cross-Linking Reagent to Advance In Vivo Cross-Linking Mass Spectrometry. *Angew Chem Int Ed Engl.* 2022 Mar 14;61(12):e202113937. doi: 10.1002/anie.202113937. Epub 2022 Jan 27. PMID: 34927332; PMCID: PMC9303249.

SI page 29: Please provide the type of sonication device used.

We have used a UP100H Ultrasonic Processor from the company Hielscher and water bath sonication. The device type was updated in the text.

SI page 34, Figure S3: In panel A, 10 mM ASSB is not shown although mentioned in the legend. In C, the 30 mM selected as optimal are beyond the range evaluated and shown here. In F, the value for 50 uL of beads is not shown, although it is mentioned in the legend.

We apologise for the confusing legend and Figures. We tested 50uL but could not interpret the data due to issues during sample acquisition, therefore we decided to remove this sample from the experiment. 30mM Sodium ascorbate (NaAsc) was selected to ensure stable conditions during the click reaction. 20mM was shown in Figure S3C as the best working condition but not reaching fully a plateau yet. Therefore we have chosen 30mM sodium ascorbate. The recommended value by the manufacturer 100mM sodium ascorbate was also

tested against click reaction with 30mM sodium ascorbate and 5mM ASSB (Figure S3 A) and using less of sodium ascorbate resulted in better enrichment performance of DiSPASO compared to 100mM, therefore we used it in our final experiments.

Why the 10mM ASSB is not shown in Figure S3 A and B is not clear, since it is the best working concentration but we included now also the 10mM ASSB in both figures.

Typos:

Page 5, legend to Figure 2: shoes > shows

Page 9: electron spray > electrospray

SI page 25: Thermo Fischer > Thermo Fisher

SI page 30: TECEP > TCEP

SI page 34: sodium adsorbate > sodium ascorbate

We thank the reviewer for the careful check. We have changed accordingly all of these typos.

Reviewer #2 (Remarks to the Author):

The author proposed an interesting cross-linking reagents for XL-MS. Overall the cross-linker showed

good trans-membrane and enrichment properties with sufficient evidence. However, apart from minor issues, my main concern with this study is the lack of comprehensive model compound (peptide) data, which should be addressed before I can recommend the manuscript for publication.

In the study, cross-linking on only one peptide is studied prior to HeLa cell XL-MS. I recommend the authors include more peptides and test the cross-linker on peptide mixtures without and with matrix (e.g. HeLa cell lysate) to better understand, evaluate, and optimize the click chemistry.

We thank the reviewer for this comment. We indeed did not only validate the click reaction based on a single peptide but also using our model protein Cas9-Helo. This extensive evaluation of the click reaction is demonstrated in Figure S3 A-F and was carried out with three different click reagents/ strategies. The figures can be read as follows: especially in Figure S3 there are 3 colours within the figures describing the start point of a click reaction (blue), the endpoint of the click reaction with its final click product (orange) and the cleavage product after enrichment of crosslinked peptides (green). If the starting product's blue bars go down and the click end product's orange bars go up, that means that the click reaction has worked. If the blue bar disappears and the orange bar reaches its maximum, the click reaction is completed with 100%. Usually, the click reaction efficiency in our experiments is between

80-99%. This kind of experiment was carried out with ASSB as a click reagent on Cas9 in Figure S3 and with picolyl azide and Cas9 in Figure S2 and on peptide level Figure S1. Since our click reaction efficiency is very high throughout our experiments, we feel that our model systems are sufficient to be able to judge the performance before in-cell crosslinking scenarios.

In addition, there are also some minor issues:

1. More proof reading on some simple phrases such as "MS-cleavable cleavable cross-linkers"

Spelling mistakes have been corrected

2. It will be better to also include mass spectra in the supplementary data.

Mass spectra of DiSPASO is now included in the supplement and raw data are uploaded to PRIDE.

Reviewer #3 (Remarks to the Author):

The authors of this manuscript present a comprehensive study on the development and characterization of DiSPASO, a novel lysine-reactive, MS-cleavable crosslinker. The linker includes an affinity handle, and several strategies have been tested to maximize the number of identifications. The work includes testing the reagent at increasing levels of sample complexity, up to live cells, and a side-by-side comparison with a similar reagent that has already been published. The manuscript is well-structured, with detailed experimental evidence supporting both the crosslinker's effectiveness and its limitations. I appreciate the manuscript as it also discusses unsuccessful attempts, highlighting the complexity of developing a new crosslinker. I believe this manuscript makes a valuable contribution to the field of crosslinking mass spectrometry and can be accepted with the following minor revisions, provided the authors make the data publicly available (PXD056091 is not currently accessible, so I could not review the data).

We thank the reviewer for this comment and apologise for not providing the login details for revision directly in the manuscript. They have been provided to the editor but unfortunately to a later stage. The login details are now included in the manuscript.

Username: reviewer_pxd056091@ebi.ac.uk

Password: qSikSCI4pNxi

Page 2: "MS-cleavable cleavable cross-linkers" should be revised to "MS-cleavable cross-linkers."

Page 4: "shows medium membrane permeability" should be revised to "shows moderate membrane permeability."

Page 5: "If a crosslinker shoes two properties" should be corrected to "If a crosslinker shows two properties."

All typos have been changed accordingly.

Reviewer #4 (Remarks to the Author):

In this work, the authors developed new cleavable crosslinker reagents for in-cell protein crosslinking and evaluated their performance for XL-MS. As the authors also noted, the reagents developed have many challenges and do not offer any particularly noteworthy advantages compared to existing reagents. Although XL-MS reagents with novel molecular structures may provide insights into future development and optimization in XL-MS applications, the current study is too preliminary to be published in Communications Chemistry, where high academic impact and novelty are required. Therefore, I recommend that this manuscript should be submitted to other journals more specific to the field of mass spectrometry.

Minor comments:

1. There are many typos in the manuscript. For example, -MS-cleavable cleavable cross-linkers -an alkyne-based click chemistry, not "alkene"

Typos were changed accordingly

2. Figure 3: There is no nitrogen atom on the benzene ring of Picoryl-Azide (3). Was changed accordingly

3. Figure 3: The structure of the BAPD (8) eluted from the beads using a reducing agent is incorrect. It should not be cleaved at the amide bond.

We thank the reviewer for this hint. Indeed the structure displayed is only a version of the crosslinker after a neutral loss during fragmentation and this was not meant to be displayed in this figure. Nevertheless, the masses used for the data analysis throughout the DAAB experiment were according to the now revised structure (8) in Figure 3C. The results of DiSPASO remain low regarding the crosslinking yield.

REVIEWERS' COMMENTS:

Reviewer #1 (Remarks to the Author):

The authors have addressed most of my comments appropriately in this revised version. I am still not convinced with the sentence in lines 285-288:

"The strong binding (free binding energy of -41.17 kcal/mol⁴⁷, K_d of $\sim 10^{-30}$ M) of the biotin group to streptavidin beads can be fully exhausted, while binding efficiency can be granted due to the reduction of the disulfide bond."

The binding can be exhausted - exhausted means tired, maybe also depleted in this context. However, I think what the authors mean here is that the strong binding ensures quantitative capture of the compound(s) containing a biotin group.

Binding efficiency can be granted due to the reduction of the disulfide bond - what does binding EFFICIENCY have to do with reduction of the disulfide bond? Maybe the authors meant reversibility of the binding?

We thank the reviewer for pinpointing again to this sentence, we have revised this sentence as follows:

"Strong binding (free binding energy of -41.17 kcal/mol [47], K_d of $\sim 10^{-30}$ M) of the biotin group to streptavidin beads allows efficient capture of labeled peptides, while elution efficiency can be achieved due to reduction of the disulfide bond."

Reviewer #2 (Remarks to the Author):

The author presented an interesting strategy and honestly reported the process for developing a new reagent for cross-linking mass spectrometry. The author to some extent addressed my concern. However, it will be better to optimize model compounds with or without the context of cell lysate to pinpoint the reason for low cross-linking efficiency.

Nevertheless, this study is well-designed and indeed every MS cross-linking reagents inevitably requires extensive optimization and lengthy, continuous efforts. Sharing such knowledge and process is important for scientific communication. In this context, I can recommend this manuscript to be published.

We fully agree with the reviewer's concern, and we would like to draw the attention in this regard to supplemental figure S7. In this figure we show our results of crosslinked *E. coli* ribosome with DiSPASO in the context of a spike in experiment (1:100), meaning that the crosslinked ribosome was spiked in in a complex HEK cell lysate background. The first blue bar represents only the crosslinked ribosome without the HEK cell lysate background and the green bar represents the enrichment of crosslinked peptides with the ASSB strategy from a complex background. We believe that the low number of crosslinks after enrichment indicates already hurdles that we were facing with the DiSPASO crosslinker for in-cell crosslinking experiments.

Reviewer #3 (Remarks to the Author):

The authors have satisfactorily addressed all the points raised during the review process, thereby improving the manuscript. I believe it is now suitable for publication in Communications Chemistry.